# Optogenetic engineering to probe the molecular choreography of STIM1-mediated cell signaling

Guolin Ma [1,7], Lian He[1,7], Shuzhong Liu[1,2,7], Jiansheng Xie[1,3], Zixian Huang[1,4], Ji Jing[1], Yi-Tsang Lee[1], Rui Wang[1], Hesheng Luo[2], Weidong Han[3✉], Yun Huang [5✉] & Yubin Zhou[1,6✉]

Genetically encoded photoswitches have enabled spatial and temporal control of cellular events to achieve tailored functions in living cells, but their applications to probe the structure-function relations of signaling proteins are still underexplored. We illustrate herein the incorporation of various blue light-responsive photoreceptors into modular domains of the stromal interaction molecule 1 (STIM1) to manipulate protein activity and faithfully recapitulate STIM1-mediated signaling events. Capitalizing on these optogenetic tools, we identify the molecular determinants required to mediate protein oligomerization, intramolecular conformational switch, and protein-target interactions. In parallel, we have applied these synthetic devices to enable light-inducible gating of calcium channels, conformational switch, dynamic protein-microtubule interactions and assembly of membrane contact sites in a reversible manner. Our optogenetic engineering approach can be broadly applied to aid the mechanistic dissection of cell signaling, as well as non-invasive interrogation of physiological processes with high precision.

[1] Center for Translational Cancer Research, Institute of Biosciences and Technology, Texas A&M University, Houston, TX 77030, USA. [2] Department of Gastroenterology, Key Laboratory of Hubei Province for Digestive System Disease, Renmin Hospital of Wuhan University, Wuhan 430060, China. [3] Department of Medical Oncology, Laboratory of Cancer Biology, Institute of Clinical Science, Sir Run Run Shaw Hospital, College of Medicine, Zhejiang University, Hangzhou, Zhejiang, China. [4] Department of Oral and Maxillofacial Surgery, Sun Yat-sen Memorial Hospital, Sun Yat-sen University, Guangzhou, Guangdong 510120, China. [5] Center for Epigenetics and Disease Prevention, Institute of Biosciences and Technology, Texas A&M University, Houston, TX 77030, USA. [6] Department of Translational Medical Sciences, College of Medicine, Texas A&M University, Houston, TX 77030, USA. [7] These authors contributed equally: Guolin Ma, Lian He, Shuzhong Liu. ✉email: Hanwd@zju.edu.cn; yun.huang@tamu.edu; yubinzhou@tamu.edu

ligand-induced conformational changes, self-oligomerization, and protein–protein interactions are among the most common strategies employed by proteins to initiate or terminate cell signaling[1,2]. Nature has evolved a wide variety of photosensory domains that can be directly utilized or further engineered as optogenetic tools to mimic these signaling processes with a simple flash of light[3,4]. Compared with conventional biophysical and biochemical methods, the optogenetic approach rivals by non-invasiveness, reversibility, and high spatiotemporal precision[4]. Over the past decade, we have witnessed an explosion of optogenetic applications in multiple fields, including neuroscience, immunology, cell biology, and developmental biology[3–6]. While most studies are centered on applying optogenetics to control biological systems at the cellular and system levels, the potential of optogenetics to aid the mechanistic dissection of protein behaviors per se remains disproportionately underexplored.

Store-operated calcium (Ca2+) entry (SOCE), as notably exemplified by the Ca2+ release-activated Ca2+ (CRAC) channel composed of stromal interaction molecule 1 (STIM1) and ORAI1, constitutes an ideal two-component system to dissect cell signaling[7–11] (Fig. 1). Under the resting condition with a fully replenished intracellular Ca2+ store, STIM1 as an endoplasmic reticulum (ER)-resident Ca2+ sensor protein is evenly distributed across the ER network. Because its cytosolic region contains an S/TxIP motif, STIM1 constantly tracks the plus ends of microtubules (MTs) via its interaction with the end-binding protein 1 (EB1)[12]. Upon ligand or antigen-triggered Ca2+ store depletion in the endoplasmic reticulum (ER), Ca2+ dissociates from the luminal EF-SAM domain of STIM1 to initiate conformational changes with subsequent oligomerization of the luminal domain[13]. Next, the luminal signal is transmitted via the single transmembrane (TM) domain of STIM1 toward the cytoplasmic domain (STIM1ct)[14]. The close apposition of the N-termini of STIM1ct dimer is believed to overcome the intramolecular autoinhibition mediated by its coiled coil 1 (CC1) and the STIM-ORAI activating domain (SOAR or CAD), thereby triggering conformational changes to expose SOAR and the polybasic (PB) C-tail[14,15]. Activated STIM1 proteins further multimerize and migrate toward the plasma membrane (PM) to physically engage and activate the ORAI1 Ca2+ channels. This process is greatly facilitated by the interaction of STIM1-PB with negatively charged PM-resident phosphoinositides (PIPs), as well as the physical association between STIM1-SOAR and the intracellular regions of ORAI1[16]. Sustained Ca2+ influx through ORAI1 channels is required to activate the downstream Ca2+-responsive transcription factor, the nuclear factor of activated-T cells (NFAT). CRAC channel activation involves STIM1 self-oligomerization, conformational switch and its interaction with PM-resident phospholipids and the PM-embedded ORAI1, thereby providing an excellent system to demonstrate the high potential of optogenetics in dissecting protein actions.

Here, we report the design of a series of synthetic optogenetic tools to probe the structure–function relationship of the CRAC channel. By engineering photosensitivity into modular domains of STIM1, we reconstructed these key molecular steps involved in the activation of SOCE in mammalian cells. The optogenetic toolkit derived from engineered STIM1 could find broad applications by permitting light-inducible gating of Ca2+ channels, controlling dynamic protein–microtubule (MT) interactions, and reversibly manipulating ER–PM membrane contact sites (MCSs) in living cells in real time. Mechanistically, STIM1-based optogenetic tools enable us to identify key molecular determinants that govern STIM1 oligomerization and conformational switch. Our tool can be further adapted into a high-throughput format to rapidly screen mutations in STIM1 that might decouple ORAI

channel-binding from channel-gating, and to aid the functional characterization of a panel of cancer-associated STIM1 mutations. Although our study presented herein is focused on STIM1, similar optogenetic engineering approaches can be extended to probe structure–function relations of other signaling molecules, thereby achieving remote control over various physiological processes with high spatiotemporal resolution.

## Results

**Optical crosslinking to trigger STIM1-gated Ca2+ influx.** In the resting condition, STIM1 is maintained in an inactive configuration by its Ca2+ loaded luminal domain and the autoinhibitory cytoplasmic domain[13,14]. Chemical-induced dimerization of the luminal domain or crosslinking at the N-terminus of STIM1ct has been previously shown to activate STIM1 and Ca2+ influx via endogenous ORAI channels[17,18], thereby mimicking the functional consequence of store-depletion-induced multimerization of the luminal EF-SAM under physiological scenarios (Fig. 1a, b). This prompted us to mimic the initial steps of STIM1 activation by utilizing an optical dimerizer made of iLID (LOV2-ssrA) and sspB[19], which undergoes blue light-dependent heterodimerization within dozens of seconds. In HeLa cells co-expressing two hybrid constructs made of iLID- or sspB-fused STIM1ct (Fig. 1c), we observed notable light-triggered Ca2+ influx, as reflected by a rapid increase in the fluorescence signal of a red genetically encoded Ca2+ indicator (GECI), R-GECO1.2 ($t_{1/2, on}$ = 28.5 ± 3.2 s (mean ± s.e.m.); Fig. 1d, e). After withdrawal of blue light, the Ca2+ signals returned to the basal level with a deactivation half-life of 48.6 ± 5.4 s (Fig. 1e), clearly attesting to the full reversibility of this synthetic system. The similar light-inducible Ca2+ entry across the plasma membrane could be recapitulated by using another optical dimerizer comprising the photolyase-homology domain (PHR) of cryptochrome 2 (CRY2; aa 1–498) and its binding partner CIBN (the N-terminal domain of CIB1, aa 1–170) from *Arabidopsis thaliana*[20] (Fig. 1f). CRY2 is known to not only undergo light-dependent homo-oligomerization[21], but also exhibit light-inducible heterodimerization with CIBN[20]. When co-expressed in mammalian cells, CRY2PHR acted as a photo-crosslinker to activate CIBN-STIM1ct, as evidenced by its cytosol-to-PM translocation to interact with PM-resident YFP-ORAI1 following light stimulation (Fig. 1g and Supplementary Fig. 1). Again, we observed reversible Ca2+ influx in response to repeated light-dark cycles of stimulation ($t_{1/2, on}$ = 23.4 ± 2.6 s and $t_{1/2, off}$ = 153.0 ± 26.2 s; Fig. 1h, and Supplementary Movie 1).

Next, we set out to further mimic STIM1 activation and puncta formation in a membrane-constrained environment as seen with the native STIM1 embedded in the ER membrane. To test this idea, we generate our first-generation constructs by fusing CRY2 downstream of the ER-targeting signal peptide or replacing the luminal EF-SAM domain with CRY2 (Supplementary Fig. 2). However, our initial trials failed since none of these constructs led to light-inducible Ca2+ influx (Supplementary Fig. 2b). This is unlikely due to damages to STIM1 modules because some of the constructs still responded well to store depletion (Supplementary Fig. 2c). We reasoned that the oxidative environment within the ER lumen (−118 mV)[22] might prohibit the photoactivation of cryptochromes, which requires a more reducing redox potential (−143 to −153 mV)[23,24] to complete the photocycle. Indeed, when tethered to the cytosolic side of ER membranes, CRY2 could be readily photoactivated to form oligomers along the ER tubules (Supplementary Fig. 2a). Therefore, we generated our second-generation hybrid constructs encoding ER-tethered CRY2-STIM1ct (Fig. 1i) or iLId/sspB-STIM1ct (Supplementary Fig. 3a, b), with all modules facing toward the cytosolic side of the

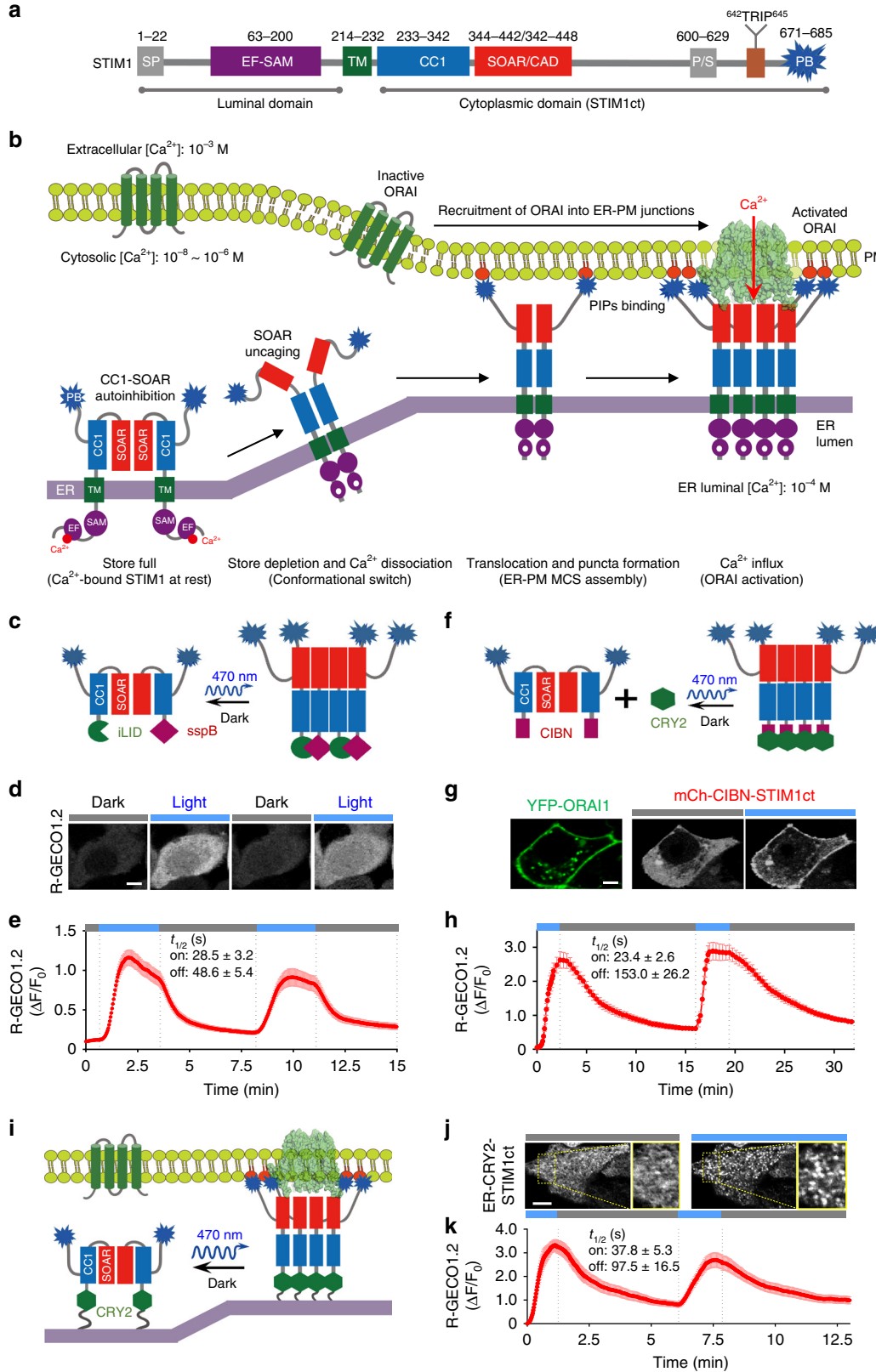

ER membrane. After blue light illumination, ER-tethered CRY2-STIM1ct formed puncta-like structure (Fig. 1j), accompanied with Ca²⁺ influx ($t_{1/2, on} = 37.8 \pm 5.3$ s and $t_{1/2, off} = 97.5 \pm 16.5$ s; Fig. 1k) and subsequent nuclear translocation of the downstream effector, NFAT (Supplementary Fig 3c, d). Likewise, ER-anchored iLId/sspB-STIM1ct was able to elicit Ca²⁺ influx in response to

light stimulation (Supplementary Fig. 3b). Together, light-induced close apposition of the N-terminus of STIM1ct (in the CC1 region), much like store depletion-induced oligomerization of the luminal domain and TM re-organization, is sufficient to trigger STIM1 activation with ultimate Ca²⁺ influx through ORAI channels. Furthermore, upon the withdrawal of light

**Fig. 1 Optogenetically engineered STIM1 permits light-switchable activation of ORAI Ca$^{2+}$ channels.** Photostimulation was applied at 470 nm (4.0 mW/cm$^2$). Data were shown as mean ± sem. Scale bar, 5 µm. **a** Domain architecture of the human STIM1. SP, signal peptide; EF-SAM, EF-hand and sterile alpha-motif; TM, transmembrane domain; CC1, coiled-coil domain 1; SOAR, STIM-Orai activating region; P/S, proline/serine-rich region; TRIP, the S/TxIP microtubule-binding motif; PB, polybasic tail. **b** Schematic of STIM1–ORAI1 coupling at the ER-PM junction that mediates store-operated Ca$^{2+}$ entry. **c–e** Use of the iLID-sspB optical dimerizer to trigger STIM1ct activation and Ca$^{2+}$ influx through endogenous ORAI channels. **c** Schematic of the design. iLID or sspB was fused to the N-terminus of STIM1ct at residue 233. **d** Confocal images showing photoswitchable Ca$^{2+}$ influx in HeLa cells co-transfected with a red Ca$^{2+}$ sensor (R-GECO 1.2) and the iLID/sspB fused STIM1ct chimeras. Cells were exposed to two repeated dark-light cycles. **e** Quantitative analysis of Ca$^{2+}$ signals in response to repeated photostimulation ($n = 40$ cells from three independent experiments). The half-lives ($t_{1/2}$) of on and off kinetics were fitted with one phase exponential decay ("±" means 95% confidence interval). **f–h** Use of the CRY2-CIBN optical dimerizer to photo-activate STIM1ct and Ca$^{2+}$ influx. **f** Schematic of the design. CRY2 was used to photo-crosslink CIBN-STIM1ct and trigger STIM1ct activation to induce Ca$^{2+}$ entry. **g** Confocal images showing light-induced co-localization of mCherry (mCh)-tagged CIBN-STIM1ct with YFP-ORAI1 in HeLa cells. **h** Reversible Ca$^{2+}$ responses monitored by R-GECO 1.2 ($n = 30$ cells). Blue bar, photostimulation at 470 nm with a power density of 4 mW/cm$^2$. **i–k** ER-tethered CRY2-STIM1ct mimics STIM1 puncta formation at ER–PM junctions to evoke localized Ca$^{2+}$ influx. **i** Schematic of the design. **j** Confocal images illustrating light-induced clustering of ER-resident CRY2-STIM1ct at the footprint of HeLa cells. Enlarged views of the boxed regions were shown on the right. **k** Cytosolic Ca$^{2+}$ signals reported by R-GECO1.2 in HeLa cells subjected to two repeated dark-light cycles ($n = 30$).

stimulation, the optogenetic module undergoes dissociation, a process that resembles the binding of Ca$^{2+}$ to the luminal EF-SAM domain to drive the de-oligomerization of STIM1 upon ER Ca$^{2+}$ store refilling[13,25]. In the absence of pro-oligomerization signals, STIM1ct will dissociate from ORAI and adopt an inactive conformation via intramolecular CC1–SOAR trapping. We speculate that forced separation of the juxtamembrane ends of CC1 might be sufficient to bring STIM1 back to its resting configuration, even in the absence of other ancillary proteins.

**Optogenetic mimicry of CC1–SOAR mediated autoinhibition.** The interaction between CC1 and SOAR is required for STIM1 autoinhibition and to keep STIM1 inactive at the resting state[14,26,27]. We confirmed this by splitting the full-length STIM1 into two components at position 342, and appended each part with a different fluorescent protein tag (Fig. 2a). At rest, Part I (STIM1$_{1-342}$-YFP) bearing the CC1 region was localized to ER and Part II (mCherry-STIM1$_{343-685}$) with the SOAR domain also displayed an ER-like distribution, implying a physical interaction between CC1 and the SOAR-containing cytosolic fragment (Fig. 2b, upper panel). Upon passive Ca$^{2+}$ store depletion induced by thapsigargin (TG), we observed an immediate release of Part II from ER toward the cytosol with discernible PM-like decoration (Fig. 2b, lower panel), likely owing to its association with endogenous ORAI channels. Because of the intramolecular CC1–SOAR interaction-mediated autoinhibition[14], STIM1ct showed an even distribution in the cytosol regardless of the filling status of the ER Ca$^{2+}$ store (Supplementary Fig. 4a, b) in cells co-expressing Part I, STIM1$_{1-342}$-YFP. However, after we introduced L258G in CC1 to abrogate the intramolecular CC1–SOAR association *in cis*[14], we observed an ER-like distribution of STIM1ct in cells co-expressing STIM1$_{1-342}$-YFP, suggesting an *in trans* CC1–SOAR interaction between STIM1ct and Part I at rest. Upon store depletion, ER-bound STIM1ct dispersed into the cytosol due to disruption of CC1–SOAR interaction *in trans* (Supplementary Fig. 4b). These data suggest that store depletion-induced conformational switch in STIM1ct or mutation-induced disruption of the CC1–SOAR interaction can overcome STIM1 autoinhibition to unleash the minimal ORAI-activating fragment, SOAR.

The above finding motivated us to explore an optogenetic approach to reconstruct the conformational switching step during STIM1 activation. We resorted to the light-oxygen-voltage domain 2 (LOV2; aa 404–546) from *Avena sativa* phototropin 1 because it undergoes allosteric conformational changes upon blue light stimulation[28]. We reasoned that the CC1 region could be replaced by LOV2 to impose steric hindrance to the downstream SOAR, thereby caging the SOAR-containing STIM1ct fragments

(e.g., STIM1$_{336-486}$ and STIM1$_{336-685}$) in the dark as CC1 did at the rest condition. Upon light stimulation with the ensuing unfolding of the C-terminal Jα to uncage the fused effector domain, we anticipated that SOAR would restore its function to interact with ER-anchored CC1 or PM-resident ORAI1 (Fig. 2c). Indeed, we observed a light-dependent recruitment of cytosolic LOV2-SOAR (STIM1$_{336-486}$) toward the ER membrane in HeLa cells co-transfected with the CC1-bearing STIM1$_{1-342}$-CFP (Part I; Fig. 2d, e), or translocation toward PM in HeLa cells co-expressing ORAI1-YFP (Fig. 2f, g, upper panels). This process could be reversibly repeated when transfected cells were subjected to multiple light-dark cycles ($t_{1/2, \text{ on}} = 7.2 \pm 1.2$ s; $t_{1/2, \text{ off}} = 28.7 \pm 4.6$ s; Fig. 2e). Compared with LOV2-SOAR, LOV2 could not fully cage the longer STIM1$_{336-685}$ fragment, thus leading to a higher basal level of intracellular Ca$^{2+}$ and partial ORAI1 binding even in the dark (Supplementary Fig. 5a–e). Upon blue light stimulation, LOV2-STIM1$_{336-685}$ evoked Ca$^{2+}$ influx and underwent translocation toward PM to colocalize with ORAI1 (Supplementary Fig. 5b–e), but showed less colocalization with STIM1$_{1-342}$-YFP (Supplementary Fig. 5f–h). As discussed below, other structural elements downstream of the SOAR domain, such as the TRIP and PB motifs, likely exert additional forces to facilitate STIM1$_{336-685}$ moving toward the PM, rather than being anchored toward CC1.

The LOV2-SOAR chimera provided a unique opportunity for us to estimate the relative binding strengths of SOAR with PM-localized ORAI1 over SOAR with ER-localized CC1 in living cells (Fig. 2c), which remained difficult to be addressed with conventional biophysical and biochemical approaches. To test this, we co-transfected HeLa cells with mCh-LOV2-SOAR, along with STIM1$_{1-342}$-CFP-T2A-YFP-ORAI1 to ensure a near 1:1 expression of both the ER- and PM-resident components. In the dark, mCh-LOV2-SOAR mainly distributed in the cytosol without noticeable co-localization with ER or PM (Fig. 2f; top panel), indicating that the SOAR domain was tightly caged by LOV2. After blue light illumination, the majority of mCh-LOV2-SOAR translocated to the ER, but not to the PM (Fig. 2f, g, bottom panel), suggesting a tighter interaction between ER-anchored CC1 and SOAR. By contrast, for cells without over-expression of the ER-resident STIM1$_{1-342}$-CFP, photo-excited mCh-LOV2-SOAR mainly distributed around PM due to its interaction with ORAI1 (Fig. 2f, g, top panel). Functionally, HeLa cells co-expressing LOV2-SOAR and ORAI1 showed a boost in light-induced Ca$^{2+}$ influx compared to cells only transfected with LOV2-SOAR (Fig. 2h). By comparison, cells co-expressing LOV2-SOAR and STIM1$_{1-342}$ exhibited a significant decline in light-induced Ca$^{2+}$ response (Fig. 2h). Evidently, the SOAR domain alone has a preference to interact with CC1 rather than to engage ORAI1 in the plasma membrane under an ideal near 1:1

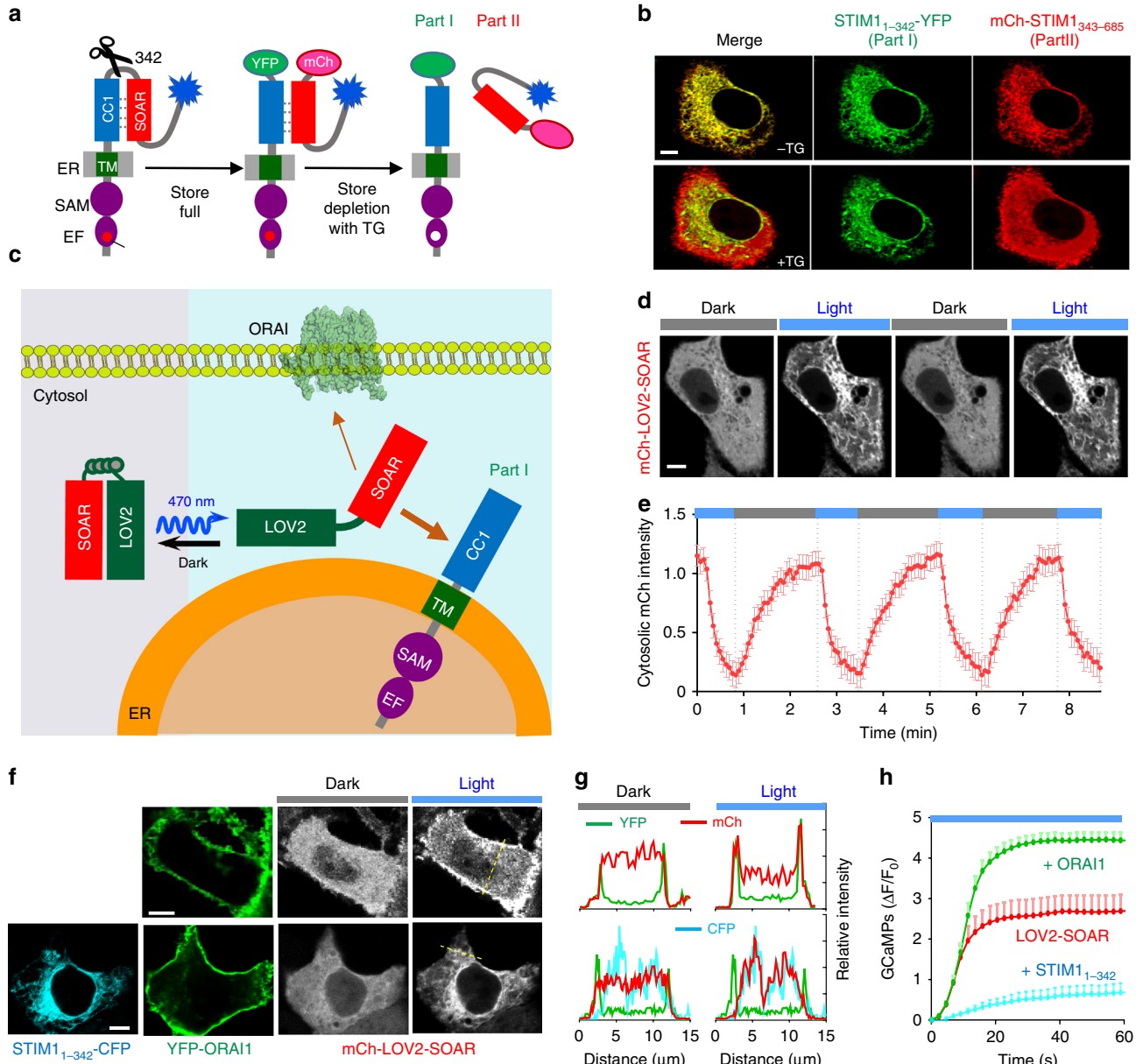

**Fig. 2 An engineered LOV2-SOAR chimera to mimic STIM1ct autoinhibition mediated by CC1–SOAR interaction.** Data were shown as mean ± sem. Scale bar, 5 μm. **a** Design of a split STIM1 molecule (at residue 342) to monitor CC1–SOAR interaction *in trans* at real time. CC1–SOAR maintains STIM1ct in an inactive configuration at rest. As a result, Part II (mCh-STIM1$_{342-685}$) tightly docks to the ER-resident Part I (STIM1$_{1-342}$-YFP) when the store remains full. Upon store depletion, structural changes propagate toward the CC1 region to weaken its association with SOAR, thereby leading to the cytosolic dispersion of Part II as shown in panel **b**. **b** Confocal images showing the distribution of split STIM1 molecules (green, STIM1$_{1-342}$-YFP; red, mCh-STIM1$_{343-685}$) before and after thapsigargin (TG)-induced store depletion in HeLa cells. **c** Schematic illustrating the design of a LOV2-SOAR (STIM1$_{336-486}$) chimera to mimic the CC1–SOAR interaction that locks STIM1ct in an inactive state. CC1 is replaced by LOV2 (light-oxygen-voltage domain 2) to tightly cage SOAR in the dark. Upon blue light stimulation, the Jα helix becomes disordered to uncage SOAR, thereby restoring its activity to engage and gate ORAI channels. If the ER-resident Part I (STIM1$_{1-342}$-YFP) and PM-embedded ORAI1 are co-expressed, LOV2-SOAR can be used to determine the relative binding strength of SOAR toward ER-anchored CC1 or PM-resident ORAI1 channels. **d** Light-inducible cytosol-to-ER translocation of mCh-LOV2-SOAR in HEK293 cells co-transfected with Part I as shown in panel **c**. **e** Quantification of cytosolic mCherry signals (images in panel **d**) following three repeated light-dark cycles (*n* = 18 cells). **f–h** Comparison of the relative strength of SOAR-CC1 and SOAR-ORAI1 interactions. **f** Top: Light-induced cytosol-to-PM translocation of mCh-LOV2-SOAR (gray) observed in HEK293 cells co-transfected with YFP-ORAI1 (green). Bottom: Confocal images of HEK293 cells co-expressing mCh-LOV2-SOAR (gray), YFP-ORAI (green) and STIM1$_{1-342}$-CFP (cyan). In the dark, mCh-LOV2-SOAR was evenly distributed in the cytosol. Upon photostimulation, mCh-LOV2-SOAR preferred to translocate toward ER membrane but not to PM. **g** The fluorescence intensities (YFP, green; mCh, red; CFP, cyan) across the dashed line were plotted to evaluate the degree of colocalization. **h** Light-induced Ca$^{2+}$ response curves (quantified by GCaMP6s) in HEK293 cells transfected with LOV2-SOAR (red), LOV2-SOAR + ORAI1 (green) or LOV2-SOAR + Part I (STIM1$_{1-342}$; blue). *n* = 30 cells.

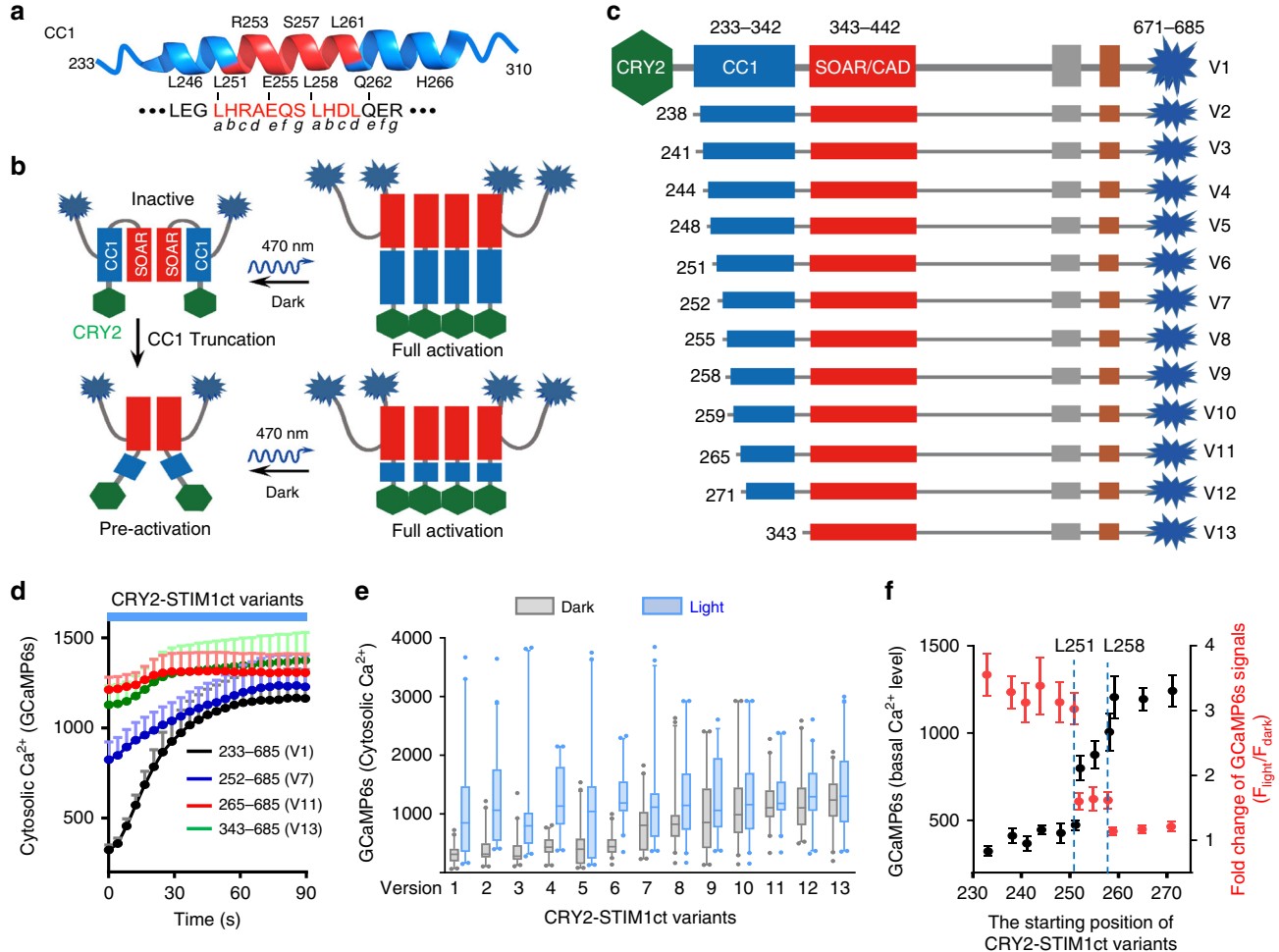

**Fig. 3 Optogenetic mapping of the CC1–SOAR contact interface that mediates STIM1ct autoinhibition.** Data were shown as mean ± sem. Scale bar, 5 μm. **a** The 3D structure of the juxta-ER membrane region of CC1 (selected residues 241–271; PDB entry: 4O9B), with the predicted helical heptad repeats indicated below the cartoon. **b**, **c** Summary of the anticipated outcomes in HeLa cells transfected with the full-length or truncated STIM1ct variants. CRY2-fusion constructs were depicted in panel **c**. **d**–**f** Light-triggered $Ca^{2+}$ responses for the indicated CRY2-STIM1ct variants. **d** GCaMP6s was used to report the basal level of $Ca^{2+}$ and light-induced changes of $Ca^{2+}$ influx in HeLa cells. **e** Box-whisker plots showing the GCaMP6s signals before (gray) and after photostimulation (blue) for the indicated CRY2-STIM1ct variants. Data were presented as median and the interquartile range with 5–95 percentile distribution. **f** The basal $Ca^{2+}$ levels (left Y axis, black dots) and fold-change of $Ca^{2+}$ signals (right $Y$ axis, red dots) plotted against the starting residues of tested CRY2-STIM1ct variants (X-axis). $n = 40$ cells from three independent experiments.

expression condition. Moreover, similar scenarios were visualized in HeLa cells co-expressing uncaged mCh-CAD/SOAR, ORAI1 and STIM1$_{1–342}$ at varying ratios (Supplementary Fig. 6). Taken together, these findings reinforce the conclusion that, in order to win the "tug-of-war" between CC1 and ORAI1 to elicit $Ca^{2+}$ influx, additional forces (discussed below) are needed to drive STIM1 translocation toward the PM.

**Optogenetic mapping of autoinhibitory regions within STIM1ct.** STIM1 autoinhibition is proposed to be mediated by coiled-coil interactions between CC1 and SOAR, with the SOAR-docking site being mapped to the juxtamembrane portion of CC1[17,18] but not fully resolved[14,27,29]. To further narrow down the key region within CC1 that mediates intramolecular trapping of STIM1, we fused CRY2$_{PHR}$ to a series of STIM1ct fragments with CC1 truncated at varying positions (Fig. 3a–c). If the CC1–SOAR interaction remained intact, we envisioned that engineered CRY2-STIM1ct variants should have very low basal level of $Ca^{2+}$ in the dark. Upon photo-illumination, CRY2$_{PHR}$ underwent oligomerization to disrupt CC1–SOAR interaction and subsequently trigger $Ca^{2+}$ influx (Fig. 3d). This turned out to

be the case for the hybrid variants truncated till the position of 251: we invariably observed low background activation in the dark but high fold-change of $Ca^{2+}$ signals in the lit condition (Fig. 3e, f). Therefore, residues preceding L251 does not seem to be involved in mediating the CC1–SOAR interaction. Subsequent deletion from L251 to L258 led to progressive pre-activation in the dark, accompanied with narrower dynamic ranges of light-triggered $Ca^{2+}$ response (Fig. 3d–f), implying that the predicted heptad repeat made of L251-S257 (*abcdefg*) is directly involved in SOAR-docking. Further truncations in the next heptad repeat (L258-R264) resulted in pre-activation of CRY2-STIM1ct variants even in the absence of light (Fig. 3d–f). In our previous mapping by truncating CC1 from the C-terminus, STIM1$_{1–261}$ could interact with the SOAR domain but STIM1$_{1–260}$ did not[14]. Thus, the predicted coiled-coil heptad repeat ranging from L251 to L261 in CC1 is critical for the CC1–SOAR association to keep STIM1 inactive at rest.

**Optogenetic clustering to dissect STIM1 oligomerization.** Oligomerization is crucial for STIM1-mediated SOCE activation, both at the early stage of sensing $Ca^{2+}$ fluctuation within the ER

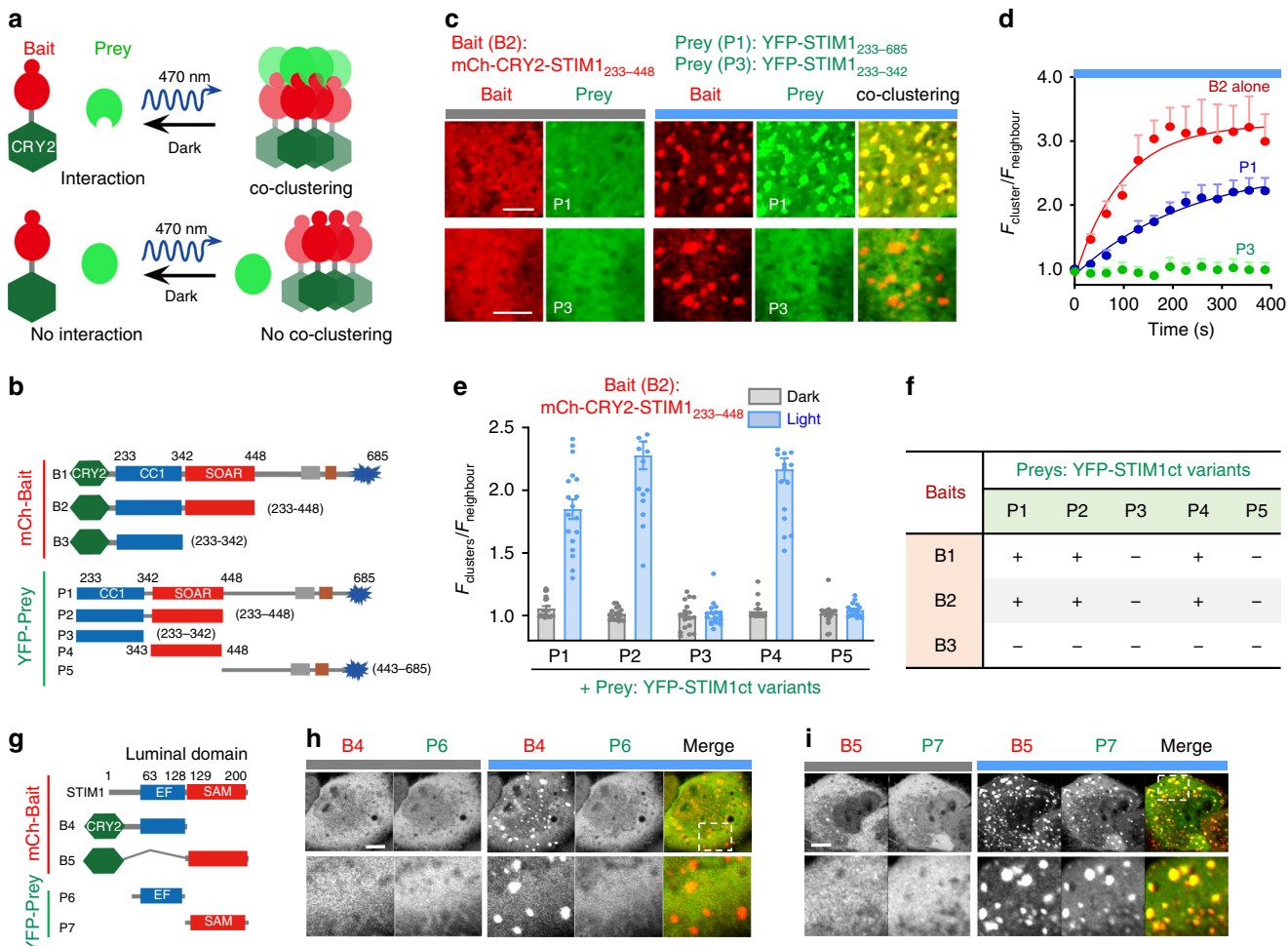

**Fig. 4 An optogenetic clustering assay to dissect key determinants of STIM1 oligomerization.** Data were shown as mean ± sem. Scale bar, 5 μm. **a** Design of an optogenetic clustering assay to examine real-time protein–protein interactions in living cells. **b** Summary of mCh-tagged baits and YFP-tagged preys used to map critical domains in STIM1ct that dictate STIM1 oligomerization. **c** Representative confocal images showing the intracellular distribution of the bait (mCh-CRY2-STIM1$_{233-448}$) and two different preys (P1—top panel, YFP-STIM1$_{233-658}$; P3—bottom panel, YFP-STIM1$_{233-342}$) before and after blue light stimulation in HeLa cells. **d** Time courses showing the kinetics of light-induced clustering (F$_{cluster}$/F$_{neighbor}$) of the bait and its co-clustering with P1 (blue), but not with P3 (green), as seen in panel **c**. $n = 18$ cells. (**e**) Quantification of the degrees of light-inducible co-clustering for the five indicated preys in HeLa cells co-transfected with the bait. $n = 18$ cells from three independent experiments. **f** Summary of the optogenetic co-clustering assay results for the bait–prey combinations shown in panel **b**. "+" means co-clustering notably observed after photo-illumination; "−" means no appreciable cluster formation before and after blue light stimulation. **g–i** Light inducible co-clustering to dissect the STIM1 luminal EF-SAM domain. **g** Schematic showing the design of bait–prey constructs. **h–i** Representative confocal images of HeLa cells co-expressing (**h**) mCh-CRY2-EF (B4, STIM1$_{32-128, \, EF-hand}$; red) with YFP-EF (P6, green), or (**i**) mCh-CRY2-SAM (B5, STIM1$_{128-200, \, SAM}$; red) with YFP-SAM (P7; red) under dark (left) and blue light (right). The selected regions (dashed boxes) were enlarged to aid visualization. Scale bar 5 μm.

lumen and at the final stage of driving STIM1 redistribution toward ER–PM junctions to gate ORAI1 channels[27,30,31]. To systematically dissect key domains involved in mediating STIM1 oligomerization in living cells, we developed a two-color optogenetic clustering assay, in which a modified CRY2 with enhanced light-inducible clustering property (CRY2$_{clust}$)[32] was fused to various mCh-tagged STIM1ct fragments to serve as the "bait" and then co-expressed with YFP-tagged "prey" proteins within the same cell (Fig. 4a, b). In the dark, both the bait and the prey stay as soluble proteins with an even distribution pattern in the cytoplasm. When exposed to blue light, CRY2$_{clust}$ drives the clustering of the bait within seconds (Supplementary Movie 2). A light-triggered co-clustering between the bait and prey proteins is anticipated if they interact with each other. Otherwise, the prey protein is anticipated to stay evenly distributed in the cytosol (Fig. 4a). Typical examples representing these two scenarios were

illustrated in Fig. 4c, d. By using mCh-CRY2-STIM1$_{233-448}$ as the bait, we observed its light-induced co-clustering with YFP-STIM1$_{233-685}$ (Fig. 4c, top panel), but not with YFP-STIM1$_{233-342}$ lacking the SOAR/CAD domain (Fig. 4c, bottom panel). We then moved on to test 15 combinations using three baits (aa 233–685 [B1], 233–448 [B2], or 233–342 [B3]) and five preys (aa 233–685 [P1], 233–448 [P2], 233–342 [P3], 343–448 [P4], or 443–685 [P5]; Fig. 4b–f, Supplementary Figs. 7–9, and Supplementary Movie 2). When the SOAR domain was not included in the bait (B3) or preys (P3 and P5), we failed to detect any noticeable co-localization (Fig. 4f), clearly attesting to an indispensable role of SOAR but not CC1 in driving STIM1 oligomerization. Moreover, if we used constructs bearing truncated SOAR domain that failed to induce Ca$^{2+}$ influx as the bait (mCh-CRY2-STIM1$_{233-430}$ or mCh-CRY2-STIM1$_{233-400}$; Supplementary Fig. 10a), we did not detect their co-clustering with the prey

YFP-STIM1$_{343-491}$ that contains intact SOAR (Supplementary Fig. 10b). Thus, the structural integrity of SOAR domain is crucial for both ORAI gating and STIM1 oligomerization.

Next, we extended the similar optogenetic clustering assay to identify regions within the ER lumen that are important for self-oligomerization of the STIM1 luminal domain (Fig. 4g). We first examined the EF-hands motif and failed to detect co-clustering of the bait–prey pair regardless of Ca$^{2+}$ concentrations (Fig. 4h and Supplementary Fig. 11). The luminal SAM domain (STIM1$_{132-200}$) tends to oligomerize when expressed in bacteria and isolated as a recombinant protein[13]. However, when expressed in HeLa cells, mCherry-tagged SAM exhibited smooth distribution in the cytosol without overt aggregation or clustering. Upon light stimulation, mCh-CRY2-SAM formed clusters within 5 min, followed by co-localization with the prey protein YFP-SAM (prey). Collectively, these results establish the ER-luminal SAM and cytosolic SOAR domains as the two major determinants driving STIM1 oligomerization during SOCE activation.

**Optogenetics aids rapid screening of STIM1 mutations.** Gain- and loss-of-function mutations in STIM1 have been reported to cause human tubular-aggregate myopathy (TAM) and severe combined immunodeficiency (SCID), respectively[33]. In addition, more cancer-associated STIM1 mutations have been reported in The Cancer Genome Atlas (TCGA) database without functional annotations[34]. There remains, therefore, a need for rapid functional characterization of disease-associated STIM1 mutations. Capitalizing on the photoswitchable CRY2-STIM1ct described above, we set out to develop an all-optical high-throughput screening (HTS) platform by combining optogenetics with mutagenesis studies. In our assays, the light-triggered cytosol-to-PM translocation of CRY2-STIM1ct mutants and Ca$^{2+}$ influx reported by GCaMP6s were used as readouts for assessing ORAI1 channel-binding and gating, respectively, at the single-cell level (Fig. 5a).

We first tested the assays by using mutations at a critical position within the SOAR domain of STIM proteins (G379 in STIM1 or the equivalent residue E470 in STIM2), which has been shown to account for the differential activation of ORAI channels[35]. The G379E mutation switched STIM1 into a STIM2-like protein by reducing the potency to activate ORAI1; whereas E470G in STIM2 is known to convert STIM2 into a STIM1-like, more potent activator of ORAI channels. We introduced similar mutations (G379E in the STIM1-SOAR domain [SOAR1]; or E470G/E470K in the STIM2-SOAR domain [SOAR2]) in the context of CRY2-STIM1ct (aa 233–685) or CRY2-STIM2ct (aa 324–833), and examined the behaviors of these hybrid variants before and after light stimulation (Fig. 5b and Supplementary Fig. 12a, b). Compared with CRY2-STIM1ct, the mutant CRY2-STIM1ct-G379E or CRY2-STIM2ct showed a very mild increase in light-elicited Ca$^{2+}$ response (Supplementary Fig. 12c) without overt cytosol-to-PM translocation (Supplementary Fig. 12d). By contrast, the CRY2-STIM2ct mutants (E470G or E470K) were able to generate pronounced Ca$^{2+}$ responses comparable to CRY2-STIM1ct upon light illumination (Supplementary Fig. 12c, d). Among these constructs, the hyperactive mutant CRY2-STIM2ct-E470K led to the strongest light-inducible Ca$^{2+}$ influx with a more rapid kinetics compared with CRY2-STIM1ct ($t_{1/2, on} = 9.3$ s vs. 38.4 s; $t_{1/2, off} = 112$ s vs. 320 s; Supplementary Fig. 12e–h). This trend was consistent with the behavior of the two full-length STIM molecules in response to store depletion under physiological conditions[35]. Therefore, the developed HTS assay could be used to faithfully report the impact of mutations on STIM-ORAI/PM interaction and STIM-mediated ORAI channel activation.

Having validated the reliability of our assays, we moved on to generate a library of STIM1 mutations through random mutagenesis in the SOAR domain, with the goal of identifying key residues that control ORAI1 binding and/or gating. Up to 600 clones of mCh-tagged CRY2-STIM1ct mutants were individually transfected in STIM1-knockout (S1-KO) HEK293 cells[36] with the co-expression of a green Ca$^{2+}$ indicator GCaMP6s and ORAI1-CFP. Seventy percent of the clones responded to blue illumination by showing differential Ca$^{2+}$ influx and/or cytosol-to-PM translocation. The distribution of selected mutants near the proposed ORAI-binding region ($^{382}$KIKKKR$^{387}$), based on their degrees of Ca$^{2+}$ responses (Y axis) and PM-targeting (X axis) before (dark dots) and after (blue dots) photostimulation, was shown in a scatter plot (Fig. 5b, c; and Supplementary Fig. 13). Interestingly, mutations introduced into the position T393 (T393X), which is located at the turn between Sα1 and Sα2 (Fig. 5b), showed diverse functional effects, as judged by their differential ORAI/PM-binding and varying activation kinetics (Fig. 5c–g and Supplementary Fig. 13). Compared with CRY2-STIM1ct (designated as WT), the mutant T393A exhibited a delayed photo-inducible activation of Ca$^{2+}$ influx ($t_{1/2, on}$: 120 ± 15 s vs. 38.4 s); whereas T393V accelerated this process with a shorter activation half-life of 13.5 ± 2.8 s (Fig. 5d). Two mutants, T393F and T393P, displayed the most striking phenotypes in an opposite manner. When expressed in cells, T393F showed a clear distribution nearby PM regardless of the presence of light (Fig. 5f, g). However, T393F led to a marked reduction in light-evoked Ca$^{2+}$ response (Fig. 5d, e and Supplementary Movie 3). When the same mutation was introduced into the full-length STIM1, we observed a similar phenotype: GFP-STIM1-T393F showed constitutive puncta formation prior to TG-induced store depletion (Fig. 5h), and caused a substantial reduction in the second peak of SOCE (Fig. 5i). In contrast, the mutant T393P did not display massive PM translocation (Fig. 5f, g), but was able to elicit strong Ca$^{2+}$ influx upon light illumination (Fig. 5e). In a third case, the mutant I383K showed light-inducible PM translocation but failed to induce Ca$^{2+}$ influx (Fig. 5e and g). In addition, both the full-length STIM1-I383K and a truncated (STIM1$_{1-448}$-I383K) variant exhibited spontaneous puncta formation in HEK293 cells depleted of ORAI1-3 (Supplementary Fig. 14), likely because of the introduction of a second polybasic PIP-binding motif to drive STIM1 activation independent of ORAI1 and the STIM1-PB domain. These examples clearly suggest that the ability of STIM1 to couple with ORAI1/PM (channel binding) does not necessarily correlate with their capability of ORAI channel gating to mediate Ca$^{2+}$ influx. Hence, the optogenetic screening platform enabled us to identify important mutations in SOAR (e.g., T393F) that might partially or fully decouple ORAI channel-binding from channel-gating.

Next, SOAR mutations found in patients were evaluated by the optogenetic HTS platform (Fig. 5j and Supplementary Fig. 15). The mutation H395Y found in lung adenocarcinoma[37] and R424W in stomach adenocarcinoma[38] showed full or partial pre-activation in the dark (Fig. 5j and Supplementary Fig. 15b, c) and were thus classified as gain-of-function mutations. R426L was reported to stabilize the quiescent status of STIM1 via forced CC1–SOAR interaction[27]. Indeed, cells expressing the R426L mutant failed to show Ca$^{2+}$ influx or PM translocation after photostimulation (Fig. 5j and Supplementary Fig. 15c). Another mutant, L402R, showed a similar loss-of-function phenotype. The immunodeficiency-related mutations[39], R426C or R429C, were not able to induce Ca$^{2+}$ influx, but retained the PM-targeting ability upon light illumination (Fig. 5j and Supplementary Fig. 15c). For other mutations, some showed no effects in Ca$^{2+}$ influx and/or PM translocation, whereas others showed a moderate reduction in function (Supplementary Fig. 15c). Taken together, the optogenetics-based HTS platform provides a fast

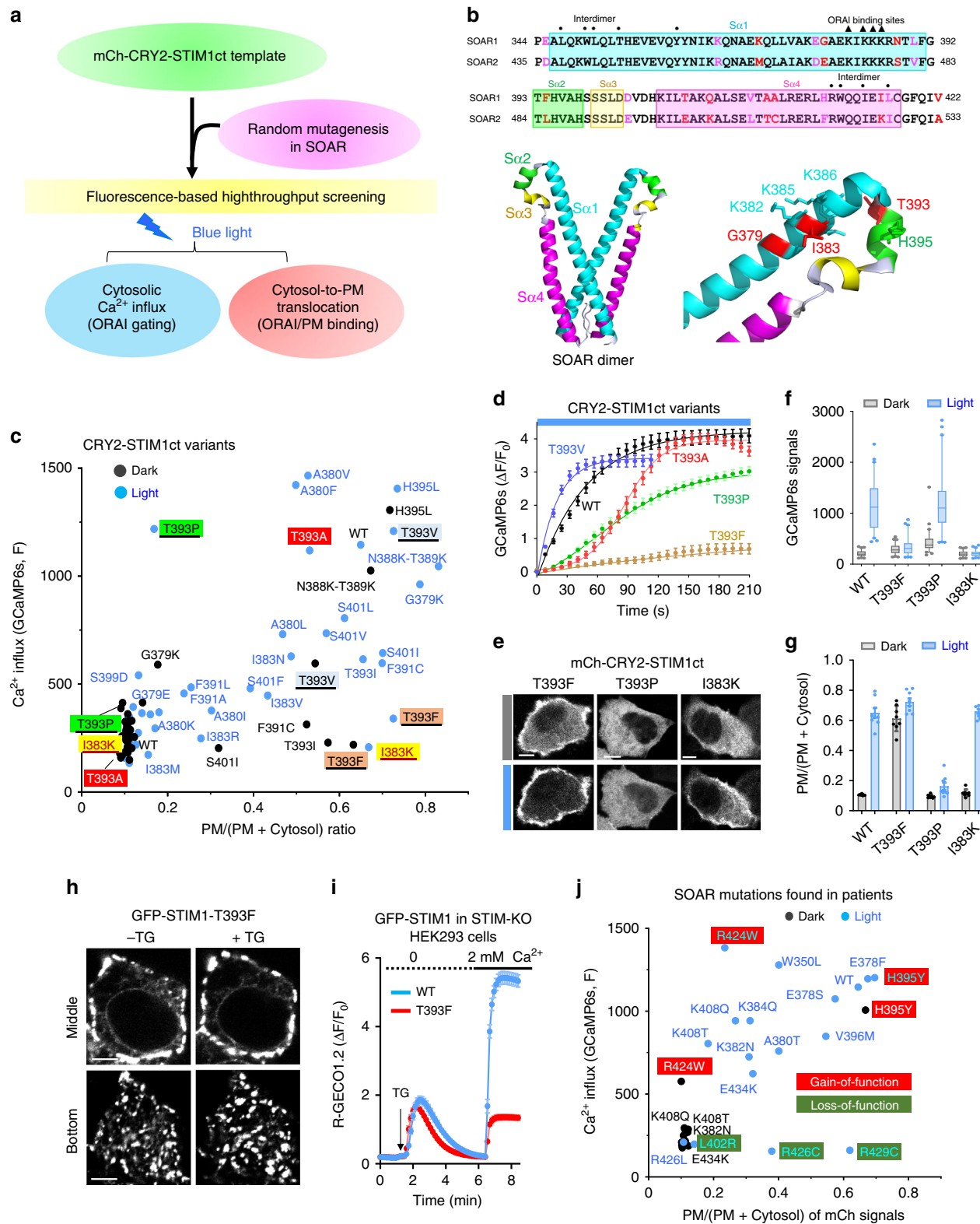

and reliable approach to characterize disease-associated STIM1 mutations.

**Optical control of STIM1–MT and STIM1–PM interactions.** Within the STIM1 cytoplasmic tail, the S/TxIP motif (TRIP; STIM1$_{642-645}$) and the polybasic domain (PB; STIM1$_{671-685}$) are functionally important (Fig. 6a). The TRIP sequence specifically interacts with EB1, a major regulator of dynamic +TIP (MT plus-end tracking proteins) interaction networks at growing microtubule ends. The positively charged PB domain is essential for recruiting activated STIM1 toward ER–PM junctions to activate ORAI channels via interaction with the negatively charged PIPs embedded in the inner half leaflet of PM[31],[40]. At rest, STIM1 is known to track the MT tips constantly. However, once activated, STIM1 stops tracking the MT plus ends and

**Fig. 5 An optogenetic platform for screening of STIM1 gain- or loss-of-function mutations.** Data were shown as mean ± sem. Scale bar, 5 μm. **a** Design of the high-throughput screening pipeline. The cytosol-to-PM translocation and $Ca^{2+}$ influx (GCaMP6s as readout) were used as two readouts. **b** Sequence alignment of human SOAR1 and SOAR2 domains and the 3D structure of SOAR1 (PDB entry: 3TEQ). Key residues at the interdimer interface or involved in ORAI1-binding were indicated by dots and triangles, respectively. Selected key residues were highlighted in the 3D structure. **c** Quantification of $Ca^{2+}$ responses (GCaMP6s) and PM translocation (mCherry signals) of selected CRY2-STIM1ct mutants before (dark dots) and after (blue dots) photostimulation. HeLa-GCaMP6s stable cells were co-transfected with each of the indicated mCh-CRY2-STIM1ct mutants and ORAI1-CFP. **d** Time courses showing the kinetics of light-induced $Ca^{2+}$ influx for WT and the indicated mCh-CRT2-STIM1ct variants. $n = 60$ cells. **e–g** Representative confocal images (**e**) and quantification of intracellular $Ca^{2+}$ signals, $n = 60$ cells. Box-whisker plots indicated the median, and the interquartile range with 5–95 percentile distribution. **f**, as well light-induced PM translocation, $n = 8$ cells (**g**), in HeLa cells expressing WT or the indicated mCh-CRY2-STIM1ct mutants. **h** Representative confocal images of HEK293 S1-KO cells expressing the GFP-tagged full-length STIM1-T393F mutant before and after TG-induced store depletion. **i** SOCE monitored by R-GECO1.2 in HEK293 S1-KO cells expressing GFP-STIM1 WT or the mutant T393F. $n = 90$ cells. **j** Summary of the degrees of $Ca^{2+}$ influx and PM translocation of cancer-associated mutations found in the SOAR domains of STIM1. HeLa cells were transfected with the indicated mCh-CRT2-STIM1ct mutants. Gain-of-function (H395Y and R424W; red) and loss-of-function (L402R, R426L/C, R429C; green) mutations were both identified. $n = 60$ cells.

migrates into ER–PM junctions to engage ORAI[9,17,25,31,40]. We envisioned that these modular motifs can be optogenetically engineered to recapitulate the tug-of-war between STIM1–MT and STIM1–PM contacts in living cells with light (Fig. 6b). To achieve this, we fused a STIM1ct fragment (aa 443–685) containing both the TRIP and PB regions with CRY2, anticipating that light-induced CRY2 clustering would increase the local avidity to boost STIM1-target interactions. When expressed in COS-7 cells, this hybrid protein stayed evenly in the cytosol in the dark, but showed simultaneous comet-like distribution and overt translocation toward the PM after blue light stimulation (Fig. 6c, d and Supplementary Movie 4). Upon removal of the C-terminal PB domain, CRY2-STIM1$_{443-670}$ only displayed a tight colocalization with EB1-GFP at the growing microtubule plus ends without any PM decoration (Fig. 6e and Supplementary Fig. 16). Deletion of the TRIP motif (STIM1$_{433-640}$; Supplementary Fig. 16) or introduction of an EB1 binding-disruptive mutation P645N (Fig. 6e, f) resulted in cluster formation due to CRY2 oligomerization, rather than MT plus end tracking, after light stimulation. Together, in the absence of PB domain, TRIP-EB1 interaction traps STIM1 to move along with EB1 at the growing MT plus ends. However, in the dual presence of TRIP and PB, oligomerized STIM1 tends to accumulate at the cell periphery to decorate PM.

A similar engineering approach was extended to the PB domain to identify key residues responsible for the PB-PM interaction (Fig. 6g and Supplementary Fig. 17a). In the dark, CRY2-PB showed an even distribution throughout the cytosol (Fig. 6g). Rapid translocation of CRY2-PB from the cytosol to PM was observed upon blue light stimulation ($t_{1/2, on}$: 7.2 ± 2.0 s; $t_{1/2, off}$: 240 ± 25 s; Supplementary Fig. 17b–d and Supplementary Movie 5). To further pinpoint key residues mediating the STIM1-phospholipid interaction, we introduced a series of mutations into CRY2-PB. A K684A mutation abolished light-induced PM translocation (Supplementary Fig. 17b), likely due to the neutralization of the positive charges required for PIPs binding. Contrariwise, introduction of additional positive charges into PB (P682K, L683K or both [PL/KK]; Supplementary Fig. 17a) enhanced the light-dependent translocation toward PM and accelerated $Ca^{2+}$ entry (Fig. 6g, h, Supplementary Figs. 17b and 18), clearly suggesting a crucial role of electrostatic interactions in PB-lipid binding.

Finally, to reconstruct STIM1-mediated ER-MT and ER–PM communications by light, we tethered CRY2-STIM1$_{443-670}$ (Fig. 6i–k) and CRY2-PB (Fig. 6l, m) toward the cytosolic side of the ER membrane, respectively. In COS-7 cells coexpressing EB1-GFP and ER-localized mCh-CRY2-STIM1$_{443-670}$, we observed the immediate EB1-STIM1$_{443-670}$ colocalization that substantially remodeled the ER network after light stimulation (Fig. 6j).

We further examined the behavior of ER-resident mCh-CRY2-STIM1$_{443-670}$ in the presence of an MT marker GFP-α-tubulin in COS-7 cells. Following blue light stimulation, mCh-CRY2-STIM1$_{443-670}$ showed clustering along the MT marked by tubulin (Fig. 6k), clearly attesting to the light-inducible formation of STIM1–MT contacts. Likewise, we monitored light-dependent changes of ER-anchored CRY2-PB variants (Fig. 6l, m). The WT CRY2-PB construct was able to photo-induce the formation of membrane contact sites between ER and PM, as reflected by the appearance of puncta-like structures at the footprint of cells (Fig. 6m, left panel). By contrast, the charge-neutralizing mutation K684A failed to bridge ER–PM junctions and only showed local clustering throughout the ER network (Fig. 6m, middle panel). The hyperactive mutant PL/KK, on the other hand, showed constitutive puncta formation in the dark (Fig. 6m, right panel). Taken together, the CRY2-based optogenetic tools allow us to dissect the molecular determinants governing STIM1–MT and STIM1–phospholipid interactions at real time under physiological conditions.

## Discussion

The prototypical CRAC channel made of ORAI1 and STIM1 serves as a major route for $Ca^{2+}$ entry in many cell types. Various optogenetic modules have been engineered into STIM1 to confer light-sensitivity to CRAC channel over the past five years[5,6,41–47]. We call these tools as genetically encoded $Ca^{2+}$ actuators (GECAs)[6], as opposed to genetically encoded $Ca^{2+}$ indicators (GECIs) that are widely used for monitoring $Ca^{2+}$ signals[48]. Two major strategies have been adopted to make GECAs: fusion with CRY2 to mimic $Ca^{2+}$-depletion induced STIM1 oligomerization or replacing CC1 with LOV2 to recapitulate intramolecular autoinhibition with the STIM1 cytoplasmic domain[5,6,41–47]. Herein, by using STIM1 as a test case, we have further exploited these optogenetic engineering approaches to study protein activities, interrogate and control cell signaling. Complementary to the existing CRAC channel-based optogenetic toolkit[41–43], we provide more examples to generate both global and localized intracellular $Ca^{2+}$ signals by harnessing the power of light. For instance, we have demonstrated the use of iLID- or CRY2-based optical heterodimerizers to photo-trigger $Ca^{2+}$ entry with varying activation and deactivation kinetics (from seconds to minutes). More importantly, by tethering the engineered STIM1ct molecules toward the cytosolic side of the ER membrane, we can mimic the formation of STIM1-like puncta at ER–PM junctions, which will likely generate $Ca^{2+}$ microdomain as seen under physiological conditions in response to store depletion[49]. A brief summary of the kinetic features of photoswitchable STIMct-based GECAs were presented in Supplementary Fig. 19, along with the

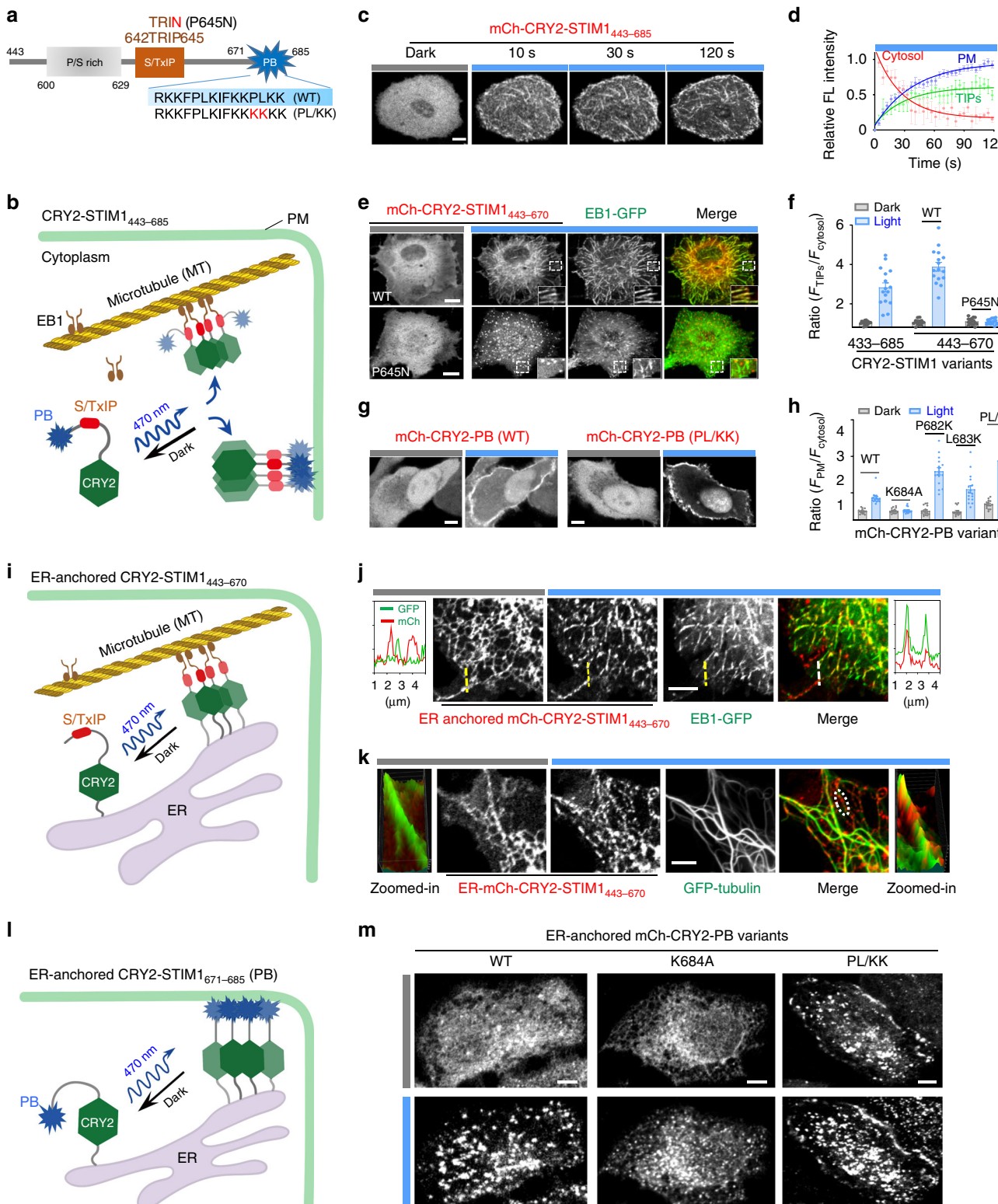

reported half-lives of ER Ca$^{2+}$ store refilling (20–70 s)[36,50,51] or STIM1 activation (20–50 s)[25,51]. Among all the constructs, iLID/sspB fused STIM1ct ($t_{1/2, \text{ on}} = 28.5 \pm 3.2$ s; $t_{1/2, \text{ off}} = 48.6 \pm 5.4$ s) seems to most closely mimic the physiological conditions. These STIM1-inspired GECAs provide multiple choices to remotely fine tune the spatiotemporal profiles of Ca$^{2+}$ signals at varying amplitudes and frequencies. Collectively, the optogenetic mimicry of STIM1ct activation through light-induced heterodimerization

or crosslinking reinforces the notion that forced apposition of the juxtamembrane coiled-coil region of STIM1 is sufficient to switch inactive STIM1 into an active configuration.

Autoinhibition as a universal mechanism used by proteins to permit temporal and spatial control of cell signals in response to extrinsic or intrinsic cues[52]. For STIM1, the autoinhibition within the cytoplasmic domain is believed to be primarily mediated by the coiled-coil clamp formed between CC1 and SOAR[13,14,27,30]. The

**Fig. 6 Dissecting STIM1–microtubule (MT) and STIM1–PM interactions with optogenetic approaches.** Data were shown as mean ± sem. Scale bar, 5 μm. **a** Diagram of the STIM1 C-terminal fragment (residues 443–685) that contains both the S/TxIP EB1-binding motif and the positively charged polybasic tail (PB) that interacts with PM-resident PIPs. Mutations used in this study to perturb STIM1ct-target interactions were highlighted in red. **b, c** Schematic (**b**) and representative confocal images (**c**) showing light-inducible bimodal distribution (tracking of MT plus ends or cytosol-to-PM translocation) of mCh-CRY2-STIM1$_{443-685}$ in COS-7 cells. **d** Time courses of light-triggered MT plus-end tracking (green) and PM translocation (blue) of mCh-CRY2-STIM1$_{443-685}$. Upon blue light illumination, cytosolic mCherry signals rapidly reduced ($t_{1/2} = 18.2 \pm 5.0$ s), accompanied with the increase of MT tip tracking ($t_{1/2} = 21.5 \pm 7.3$ s) or PM decoration ($t_{1/2} = 27.6 \pm 6.2$ s). $n = 12$ cells. **e, f** Representative confocal images (**e**) and quantifications of MT plus-end tracking (**f**) of mCh-CRY2-STIM1ct variants (top, WT; bottom, P645N in the context of STIM1$_{443-670}$) in COS-7 cells co-transfected with EB1-GFP (green). The bar graph showed the averaged values of MT tip-to-cytosol intensity ratio under dark and lit conditions. $n = 16$ cells from three independent experiments. **g, h** Representative confocal images in HeLa cells (**g**) and quantification of the cytosol-to-PM translocation (**h**) of the indicated CRY2-PB variants before and after blue light illumination. $n = 16$ cells from three independent experiments. **i–k** Schematic showing light inducible ER-MT interactions in COS-7 cells cotransfected ER-anchored mCh-CRY2-STIM1$_{443-670}$ (red) with (**j**) EB1-GFP (green) or (**k**) GFP-tubulin (green). **j** Representative confocal images showing ER morphology change following blue light illumination. The mCherry and GFP fluorescent intensities across the dashed line were plotted next to the images to indicate the degree of signal overlaps. **k** Confocal images showing the clustering of ER-resident mCh-CRY2-STIM1$_{443-670}$ along with GFP-tubulin (green) upon blue light illumination. The surface profiles of a selected area (oval) were presented to aid the visualization of subcellular distribution. **l, m** Light-inducible assembly of ER-plasma membrane contact sites (MCSs) mediated by ER-resident mCh-CRY2-PB (STIM1$_{671-685}$). **l** Schematic of the design. **m** Representative confocal images of the footprint of HeLa cells transfected with mCh-CRY2-PB variants before and after blue light illumination.

CC1–SOAR interaction *in trans* has been further visualized in living cells[14,27]. In the current study, by replacing CC1 with the LOV2 photoswitch[28] to impose steric hindrance on the SOAR domain, we have shown that light-inducible conformational switch can expose SOAR, enabling its interaction with both the ER-anchored CC1 region and the PM-embedded ORAI1 channels. Under an artificial near 1:1 expression condition, SOAR exhibited a higher affinity toward CC1, rather than the ORAI1 channels. Extrapolating this finding into a physiological scenario, additional forces are required to shift the equilibrium of STIM1 toward an activated configuration, thereby preventing the intramolecular CC1–SOAR interaction to lock STIM1 into an inactive state. The conformational switching process might be facilitated by the STIM1–PIPs interaction via the positively charged C-tail, as well as the clustering of ORAI channels to increase local avidity.

The optogenetic approach also allows us to map out a critical coiled-coil heptad repeat (L251 to L261) within CC1 that restricts SOAR from being exposed toward PM/ORAI, thus providing the molecular basis underlying STIM1ct autoinhibition. Using a light-inducible co-clustering assay, we have further identified SAM and SOAR as two critical regions responsible for self-association of the STIM1 luminal and cytoplasmic domains, respectively. Compared to conventional pulldown or immunoprecipitation-based biochemical method, the imaging-based assay allows one to assess protein–protein interactions at the single-cell level at real time under a physiologically relevant condition. In addition, since the bait protein can be oligomerized to varying degrees depending on the intensity and duration of light stimulation, it might be able to capture transient and weak interactions between the protein and target.

Capitalizing on the light-dependent oligomerization nature of CRY2, we have further faithfully recapitulated STIM1–ORAI1, STIM1–MT and STIM1–PIPs/PM associations in living cells. CRY2-STIM1ct has been further reconfigured to aid the high-throughput screening of mutations that are associated with human diseases. The CRY2-TRIP and CRY2-PB proteins, when expressed in mammalian cells, can rapidly track the MT plus ends or translocate from the cytosol toward the PM in response to light stimulation in a reversible manner. On the basis of these constructs, we could quickly map out key residues involved in contact with subcellular structures. This strategy will be very useful for probing protein–phospholipids interaction by obviating cell lysis and the isolation of proteins to reconstitute the biological process in a non-physiological artificial environment.

Worthy to note, our optogenetics-aided screening of SOAR mutants has unexpectedly revealed T393 as a hotspot crucial for ORAI-binding and gating. T393 is strategically located at the apical turn connecting Sα1 and Sα2 helices of SOAR and is also close to the proposed ORAI binding region ($^{382}$KIKKK$^{386}$). The downstream short Sα2 and Sα3 helices are known to play specific functional roles in STIM1 activation and ORAI binding or gating[39,53,54]. Mutations at T393 lead to diverse functional outcomes when using PM translocation (indirect readout for channel binding) and Ca$^{2+}$ influx (indirect indicator for channel gating) as readouts. T393F probably represents an uncommon example that could decouple channel binding from gating. T393P showed less prominent PM localization but retained the ability to gate ORAI channels, suggesting that it might act as a more potent ORAI activator. For the mutant I383K, we observed a light-dependent translocation from the cytosol to the plasma membrane, but it failed to elicit Ca$^{2+}$ influx. When we introduced this mutation into the full-length STIM1 or STIM1$_{1-448}$, we observed constitutive puncta formation at ER–PM junctions in the absence of ORAI or STIM1-PB, suggesting that the $^{382}$KIKKK$^{386}$ > KKKKK substitution at the SOAR apex region might independently promote the interaction between STIM1 and PM. We speculate that the I383K mutation might accidentally introduce a new PIP-binding motif to drive STIM1 translocation toward PM as STIM1-PB does, an interesting idea warranting further investigation in follow-on studies.

In summary, we have used STIM1 as a proof-of-concept example to illustrate how optogenetics can be applied to interrogate the structure–function relationship of a signaling protein. The optogenetic approach enables us to efficiently identify key molecular determinants governing protein oligomerization, conformational switch, autoinhibition, and protein-target interactions in a high-throughput format. The benefits of such engineering efforts are two-fold. First, the optogenetic reconstruction process can shed light to the molecular mechanisms underlying a signaling event. Second, the resultant tools (e.g., GECAs) can be further used to remotely control cell signaling with tailored applications in both basic and translational researches. For instance, the tools developed in this study can be applied to remotely control a myriad of calcium-dependent physiological processes, and to perturb the dynamics of cytoskeleton or inter-membrane contacts between subcellular organelles.

## Methods

**Molecular cloning and plasmid construction.** Plasmid construction was performed using the standard restriction enzyme digestion and ligation method. KOD Hot Start DNA polymerase was purchased from EMD Millipore (Burlington, MA,

USA) and used for PCR amplifications. Oligonucleotides were synthesized by the Integrated DNA Technologies (Coralville, IA, USA). The T4 DNA ligase kit and NEBuilder HiFi DNA Assembly Master Mix were purchased from New England BioLabs (Ipswich, MA, USA). QuikChange Multi Site-Directed Mutagenesis Kit and random mutagenesis kit was obtained from Agilent Technologies (Santa Clara, CA, USA).

Human STIM1$_{1-342}$-CFP and STIM1$_{1-342}$-YFP were generated by inserting the cDNA encoding human STIM1 (residues 1–342) into the pECFP-N1 or PEYFP-N1 vectors (Clontech; Mountain View, CA, US) between the XhoI and BamHI restriction sties. YFP/mCherry-hSTIM1 variants were prepared by amplifying the corresponding STIM1 fragments via standard PCR and then inserted the into a modified pEYFP-C1 or pmCherry-C1 vector. cDNAs encoding the full-length STIM1, ORAI and/or fluorescent proteins were inserted into pCMV6-XL5 (Origene) and pCDNA3.1(+, Invitrogen) to obtain pCMV6-XL5-GFP-STIM1 and mCherry-ORAI1[14]. Full-length STIM1 mutants were generated by using the QuikChange Site-Directed Mutagenesis Kit (Agilent). The used primers were listed on the Supplementary Table 1.

To add photosensitive domains into the cytoplasmic domain of human STIM1 (hSTIM1$_{233-685}$), we first amplified the iLID (LOV2-ssrA) and sspB components from the templates pLL7.0-Venus-iLID-Mito (Addgene; #60413) and pQE-80L-MBP-sspB-Nano (Addgene; #60409), and then inserted them into mCh-STIM1$_{233-685}$ (pmCherry-C1) or YFP-STIM1$_{233-685}$ (pEYFP-C1) with a flexible linker (SGGGGGGGG)$_3$ to obtain mCh/YFP-tagged iLID-STIM1$_{233-685}$ or sspB-STIM1$_{233-685}$. Similarly, other photosensory modules such as the PHR domain (CRY2$_{1-498}$) of *Arabidopsis thaliana* CRY2 (Addgene; #70159) and its binding partner CIBN (CIB1$_{1-180}$; Addgene; #47458) were also amplified and inserted into mCh/YFP-STIM1$_{233-685}$. For CRY2 fused STIM1 variants, STIM1 fragments with varying lengths were amplified and used to replace the STIM1$_{233-685}$ fragment in CRY2-STIM1$_{233-685}$ via the restriction enzyme digestion method. To generate CRY2 variants with enhanced clustering capabilities[32], a 9-residue peptide (ARDPPDLDN) was appended to the C-terminus of CRY2-PHR in the corresponding mCh-CRY2 fused STIM1 fragments. To construct ER-localized STIM1 variants, YFP/mCh-iLID/sspB/CRY2-taged STIM1$_{233-685}$ were amplified by standard PCR and subsequently inserted after the signal peptide and the single transmembrane domain of STIM1 in the backbone of a previously developed construct LiMETER[40,55] (pcDNA3.1-based) by using the BamHI-XhoI sites. For ER lumen localized constructs, the ER targeting sequence was inserted upstream of mCh-CRY2 fused STIM1 fragments. For LOV2 caged SOAR-containing fragments, the Rac1 was substituted in pTriEx-LOV2-Rac1 (Addgene# 22024). YFP/CFP-ORAI1 was made by inserting YFP/CFP between the BamHI and EcoRI restriction sites and human ORAI1 between EcoRI and XhoI sites in the pCDNA3.1(+) vector (Life Technologies, Carlsbad, CA, USA). The CRY2-PHR cDNA was derived from CRY2$_{PHR}$-mCh-Rho (Addgene; #42958). EB1-GFP and EGFP-alpha-tubulin were purchased from Addgene (#17234 and #12298).

To generate the SOAR mutant library, mCh-CRY2-STIM1ct in the pmCherry-C1 vector was first modified by introducing two restriction sites flanking the SOAR domain (STIM1$_{344-442}$): CTAGAA > CTCGAG to generate a XhoI site at positions 335 and 336, and CCTGGC > CCCGGG as a new XmaI site at positions 445 and 446. Error prone PCR was performed when amplifying the cDNAs encoding the SOAR fragment. Reaction conditions were optimized to yield a mutation frequency of 1–4 mutations per 1,000 base pairs according to the manufacturer's protocol by changing the PCR cycles and the amount of the template DNA. The wild-type SOAR domain in the modified plasmid of mCh-CRY2-STIM1ct (pmCherry-C1) was replaced by error-prone PCR products by taking advantage of the newly introduced restriction sites XhoI and XmaI. Plasmid DNA was isolated using the E-Z 96 FastFilter Plasmid DNA Kit (Omega Bio-tek, Inc. Norcross, GA, USA). The concentration of each plasmid was quantitated and normalized to 50 ng/μl using a 96-well UV spectrophotometer (BioTek, Winooski, VT, USA). Sanger's sequencing was performed to confirm the mutations. In parallel, some cancer-associated mutations found in patients were generated by using the QuikChange Site-Directed Mutagenesis Kit. Lentiviral vectors were generated by subcloning the GCaMP6s genes (Addgene #52228) into the LentiCas9-Blast (Addgene #52962) plasmid using standard molecular cloning procedures.

**Cell culture and transfection.** HeLa, HEK293 and COS-7 cell lines were purchased from ATCC. STIM1 knockout (S1-KO) and ORAI triple knockout (ORAI-KO) HEK293 cells were prepared by using the CRISPR/Cas9 genome editing technology with sgRNA inserted into the lentiCRISPRv2 vector (Addgene; #52961)[36]. The HeLa-GCaMP6s stable cell line was prepared by infection of HEK293 cells with lentiviruses encoding GCaMP6s. Cells were cultured at 37 °C with 5% CO$_2$ in Dulbecco's Modified Eagle medium (Sigma-Aldrich; St. Louis, MO, USA) supplemented with 10% FBS and 1% penicillin/streptomycin. For fluorescence imaging experiments, cells were seeded in 35-mm glass-bottom dishes (Cellvis, Mountain View, CA, USA). On day 2, transfection was performed using the Lipofectamine 3000 (Life Technologies; Carlsbad, CA, USA) reagent by following the manufacturer's instructions. On day 3–4, Transfected cells were mounted on the microscope stage for imaging.

**Live-cell imaging, photostimulation and image analysis.** Fluorescence imaging was performed on a Nikon Eclipse Ti-E microscope equipped with an A1R-A1

confocal module with LU-N4 laser sources (argon-ion: 405 and 488 nm; diode: 561 nm) and a live-cell culture cage (maintaining the temperate at 37 °C with 5% CO$_2$). 60 × oil or 40 × oil lens was used for high resolution imaging. To perform photostimulation with repeated pulses of dark-light cycles, an external blue light source (470 nm, 4.0 mW/cm$^2$; ThorLabs Inc., Newton, NJ, USA) or the built-in 488-nm laser source (1–5% input) was used.

To monitor light-induced Ca$^{2+}$ influx in cells expressing engineered STIM1 fragments, a red genetically encoded Ca$^{2+}$ indicator, R-GECO1.2, was co-expressed in HeLa cells. The 561-nm laser source was applied to excite R-GECO1.2 without spectral overlap with the optogenetic activation window in the range of 450–490 nm. This allowed us to monitor both the ON and OFF phases of Ca$^{2+}$ responses by simply applying dark-light cycles with the 488-nm laser source from the Nikon A1R + confocal microscope or using an external pulsed LED light (at 470 nm with a power intensity of 4.0 mW/cm$^2$). For measurements of Ca$^{2+}$ influx in HeLa cells co-expressing the green Ca$^{2+}$ indicator GCaMP6s and mCh-tagged STIM1 variants, 488-nm and 561-nm laser sources were used to excite GFP and mCherry, respectively, at an interval of 8 s. The collected images were analyzed by the NIS-Elements AR microscope imaging software (Nikon, NIS-element AR version 4.0). 40–60 cells were selected to define regions of interest (ROI) for analyzing time-lapse images of Ca$^{2+}$ influx. All experiments were repeated three times.

The optogenetic co-clustering or co-localization assay was performed in HeLa cells transiently transfected with the indicated constructs (mCh-CRY2-based baits and YFP-tagged preys; or vice versa). The defined regions of interest (ROI), such as CRY2 clusters and the neighboring cytosolic areas were measured by the ROI tool that enables quantification of the signal intensities. Next, the ratio of fluorescence intensity of cluster vs. the neighboring cytosolic mean intensity (F$_{cluster}$ / F$_{neighbor}$) was used to determine the clustering efficiency. 6–8 typical regions were selected per cell to calculated the averaged cluster-to-cytosol ratio. 10–15 cells were selected to quantify the cluster-to-cytosol ratio. To visualize the interaction of engineered STIM1 molecules with the microtubule or plasma membrane (STIM1–MT or ER–PM interactions), COS-7 cells or HeLa cells were co-transfected with the indicated constructs. The degree of colocalization was analyzed by using the Intensity Line Profile toolbox in the Nikon Elements software. All collected data were plotted by using the GraphPad Prism package (San Diego, CA, USA). To analyze the kinetics of chimeric STIM1 fragments, the apparent activation and deactivation half-lives of fluorescent signals from multiple cells were calculated by fitting the data with a single component exponential decay function. The 95% confidence interval of averaged half-lives was provided.

To observe store-operated Ca$^{2+}$ entry, intracellular Ca$^{2+}$ levels were monitored by R-GECO1.2 in S1-KO HEK293 cells co-transfected with GFP-STIM1 variants and ORAI1-CFP. 24 h after transfection, cells cultured on glass-bottom dishes were kept in a Ca$^{2+}$ free solution (107 mM NaCl, 7.2 mM KCl, 1.2 mM MgCl$_2$, 11.5 mM glucose and 20 mM HEPES–NaOH, pH 7.2) at room temperature for 30 min. 1 μM TG was used to induce store depletion. After store depletion, the incubated buffer was switched to a 2 mM Ca$^{2+}$ extracellular buffer with 107 mM NaCl, 7.2 mM KCl, 1.2 mM MgCl$_2$, 11.5 mM glucose and 20 mM HEPES–NaOH, pH 7.2. Traces shown are representative of three independent repeats with each including 40–60 cells.

**High-throughput screening of SOAR mutations.** HeLa-GCaMP6s stable cells were seeded in a glass-bottom 96-well microplate (Cellvis, Mountain View, CA, USA) at a density of 1×10$^4$ cells/well and cultured in 40 μL DMEM supplemented with 10% FBS in 5% CO$_2$ at 37 °C. 12 h later, the individual SOAR mutant from the generated library (in the backbone of mCherry-CRY2-STIM1ct) and ORAI1-CFP were co-transfected into HeLa cells by Lipofectamine 3000. 24 h posttransfection, the microplate was placed on the Nikon Eclipse Ti-E microscope stage. GCaMP6s and mCherry signals were both recorded by setting the excitation at 488 nm and 562 nm, respectively. The time-lapse images were captured by a 40X lens for every 8 s during the period of 150 s. Two fields were selected for each well to record the signals by using the add-on Nikon automation module. The 488-nm laser was used as the light source for photostimulation to elicit Ca$^{2+}$ influx and trigger the cytosol-to-PM translocation of mCherry-CRY2-STIM1ct in HeLa cells.

The data analysis was performed by using the NIS-Elements AR microscope imaging software (Nikon, NIS-element AR version 4.0). For cytosolic Ca$^{2+}$ imaging measured by GCaMP6s, 40 – 60 cells were automatically selected in one field and analyzed by defining regions of interest (ROIs). For analysis of light-induced cytosol-to-PM translocation of the mCherry signals, the "Intensity Line Profile" function in the Nikon Elements software was employed. The degree of cytosol-to-PM translocation was calculated as F$_{PM}$/(F$_{PM}$ + F$_{Cytosol}$), where F$_{PM}$ and F$_{cytosol}$ stand for the mCherry signals across the drawn line, from 10–20 cells.

**Statistical analysis and reproducibility.** Quantitative data are showed as mean ± s.e.m. unless otherwise explained. The analyzed number (n) of samples were listed for each experiment. Acquired data were analyzed in Graphpad Prism 8 and Microsoft Excel 2013.

For representative confocal images in Figs. 2b, 4h-i, 5h, 6c, 6j-k, 6m, S2a, S2c, S4b, S5a, b, S5h, S8c, S14a, b, S16 and S17b, each experiment was independently repeated at least three times with similar results.

**Reporting summary**. Further information on research design is available in the Nature Research Reporting Summary linked to this article.

## Data availability

Supplementary Data are available online. The source data underlying Figs. 1e, h, k, 2e, h, 3d-f, 4d, e, 5c, d, f, g, i, j, 6d, f, h and Supplementary Figures 1d, 2b, 3b, d, 5c-e, g, 6d, 7c, 9d, 10a, 11c, e, 12c-e, g, h, 13b-f, 15c, 17d, and 18b, c were provided as a Source Data file. The plasmids and all other data will be made available from the corresponding author upon reasonable request.

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

## Acknowledgements

This work was supported by the China Scholarship Council (to S.L. and Z.H.), the Welch Foundation (BE-1913-20190330 to Y.Z.), the American Cancer Society (RSG-16-215-01-TBE to Y.Z. and RSG-18-043-01-LIB to Y.H.), the Cancer Prevention and Research Institute of Texas (RP170660 to Y.Z.), the National Institutes of Health (R01GM112003 to Y.Z.), and the John S. Dunn Foundation (to Y.Z.).

## Author contributions

Y.Z. and G.M. conceived the ideas, designed the study and directed the work. G.M., S.L., L.H., Z.H., J.X., R.W., Y.L., and J.J. designed and generated all the plasmid constructs. G.M., L.H., and S.L. developed and characterized light-induced $Ca^{2+}$ influx, CC1–SOAR binding interface, ER-MT and ER–PM interactions. G.M. and S.L. performed the optogenetic clustering assay. G.M., S.L., L.H., W.H., H.L., Y.H. and Y.Z. analyzed data. W.H. and Y.H. provided intellectual inputs. G.M., Y.H. and Y.Z. wrote the manuscript.

## Competing interests

The authors declare no competing interests.
