## [Peer Review File · Nature Communications]

Reviewers' comments:

Reviewer #1 (Remarks to the Author):

Ma et al. provide a highly impressive study on the use light-sensitive protein domains from plants to understand the STIM1 activation mechanisms. The study is very convincing and of broad interest in the CRAC channel field. Upon reading this story several questions came to my mind, which should be addressed or commented by the authors.

In Figure 1 several engineered light-sensitive constructs are shown. Fig 1i shows light-sensitive constructs attached to the ER membrane. Another interesting approach would be if the authors link CRY2 via a TM-segment located in the ER-membrane to CC1-SOAR (in other words: CRY2 would localize in the lumen of the ER and be connected via a TM segment of CC1-SOAR located in the cytosol). This would even more mimic the wild-type STIM1 situation, just having a light-sensor instead of a Ca²⁺ sensor.

CIBN-Cry2 and Cry2-constructs show slower reversibility than iLID-sspB constructs. The authors write that their studies suggest that store-refilling is sufficient for STIM1 oligomer to monomer transition. Which of the four approaches mimics most the physiological conditions, especially in respect to time-required for store-refilling/recovery from light-activated state and associated transition of STIM1 into the resting state?

Supp Fig 1: Can you please provide a time course for NFAT translocation.

Cry2 and LOV2 attachment to STIM1 fragments have been performed by some groups. Can you please highlight and reference that and of whom basic principles have already been published.

In Figure 3 the authors present the series of CC1 truncation mutants showing that CC1-L251-L261 is critical for CC1-SOAR interaction. The enhance the reliability of this assay: Can the authors also show that truncation of part of SOAR abolishes Ca²⁺ entry, independent of the dark or light state?

Regarding the prey and bait approach in Figure 4: The authors show that the CRY2-EF domain enables light-induced clustering, but no interaction with other EF-domains. What is the effect of luminal Ca²⁺ in this case? This approach seems to be done in the cytosol with low Ca²⁺ levels.

Another interesting approach would be if the authors attach Cry2 at the NT of full-length STIM1 and investigate the clustering efficiency in dependence of the absence or presence of EF, SAM, The authors demonstrate a predominant localization of mCh-LOV2-SOAR to the ER than to the PM upon illumination with light and in the presence of STIM1 1-342 and Orai1 at a ratio of 1:1. What would happen if the ratio would be altered either via additional overexpression of STIM1 1-342 or Orai1? This could show if the ratio plays a role or not. Furthermore the 1:1 ratio is probably not physiological.

The authors describe in detail the effects of diverse single point mutations at position T393. However, the relevance of 393 remains unclear. Is it a disease related mutant? Why were other mutations not investigated in such detailed manner? Further T393 is close to the critical and prominent F394 mutant, which has not been evaluated in Fig 5c.

The assay in Fig5 e/g shows mutants with very diverse functional effects. T393F shows no Ca²⁺ entry, but strong PM localization independent of dark or light state, suggesting a defect in gating. T393P shows light-induced Ca²⁺ entry but very low PM localization – how can this be explained? I383K shows no Ca²⁺ entry, but light-induce pm-localization – is this another gating deficient effect?

Figure 5 c includes a series of unlabeled dots. Which mutants do they represent? Moreover, the WT

dot: does it represent the mCh-cry2-STIM1ct wt construct?

Also Fig 5j shows several dots without labelling. Please add labels or remove dots.

Supp Fig 7/8: Why are some mutants encircled via red rectangles?

Also the MT/PB-approach is very impressive. Still the question remains if a Cry2-STIM1-CT construct behaves in the same manner regarding localization to microtubule. Maybe the authors could provide some key experiments with their Cry2-S1-CT construct showing in the w.t. conditions and upon P645N and PL/KK mutation the MT and PM association.

The authors reported that LOV2-SOAR preferentially goes to the ER than to the PM under 1:1 S1:O1 ratio. They suggest that further domains are required for preferential recruitment to the PM. So what happens if the attached LOV2 to a fragment 344-685, thus including SOAR, TRIP and PB domains? Mutations in TRIP and PB could narrow down potential domains which allow preferential recruitment to the PM.

Minor:

Page 4: progress instead of prowess

Reviewer #2 (Remarks to the Author):

In their study, Ma et al. are utilizing multiple optogenetic modules to demonstrate that these tools, which have become a popular and powerful approach for acutely manipulating biomolecule behavior in living cells, can also drive detailed biochemical analyses of protein structure and function. The authors use STIM/Orai-mediated calcium entry as their proof-of-concept model and probe several of the molecular determinants related to auto-inhibition, oligomerization, and microtubule and plasma membrane interactions, in addition to performing some high-throughput imaging studies to link disease-related STIM1 mutations to specific functional alterations. Overall, this technically rigorous and careful study highlights an interesting new direction for the use of optogenetic actuators.

Minor points:

- 1) The authors could provide a more thorough quantification of their live-cell imaging data. Currently, only a subset of experiments are presented with quantification, mostly via calcium indicator time-courses or changes in calcium release or localization, while many results are summarized using representative images alone (e.g., Figure 4h, Supplementary Figures 3-5). Notably, very few time-courses directly tracking the relocalization/redistribution of tagged STIM1 constructs in response to illumination are shown, though it would be useful to compare the kinetics of optogenetic switching with downstream events (e.g., calcium release in Figure 1).
- 2) Looking at Supplementary Figure 10 and Supplementary Video 5, the mCh-CRY2-PB (PL/KK) construct appears to exhibit fairly strong nuclear localization at rest and to undergo striking and dynamic redistribution within the nucleus upon illumination. However, this behavior is not mentioned in the text. Could the authors please include some comment on what might be happening? Is this merely an artifact?
- 3) The sentence on page 5 starting with "Furthermore, this photo-induced process can be readily reversed..." should be clarified. From the context, it seems like the authors may be referring to spontaneous dissociation of STIM and Orai being sufficient to drive reversal in the absence of pro-dimerization signals, however the wording is somewhat awkward.

Other points:

- 4) On page 10, paragraph 2, "Fig 3f-g" and "Fig 3d-e" should refer to Figure 5.
- 5) At the top of page 11, "CRY2-baesd" should be "CRY2-based".
- 6) At the top of page 12, "SAOR" should be "SOAR".
- 7) In cartoon in Figure 2c, the LOV and SOAR domains switch colors after light stimulation. Also, in Figure 2f, the mCh channel images are show in grayscale but are described as "red" in the legend.
- 8) The legends for Figure 5e and f are inverted with respect to what is shown in the figure.
- 9) On the last page of the Supplement, "Supplementary Videos 1-4" should be "1-5".

We thank the editor for providing us a chance to improve our manuscript. We have performed the recommended experiments, and supplied new data or comments to address all the concerns raised by both reviewers. All changes made in the main text and supplementary materials were highlighted in blue.

Reviewer #1

We are thankful to the reviewer for the supportive remark that *“Ma et al. provide a highly impressive study on the use light-sensitive protein domains from plants to understand the STIM1 activation mechanisms. The study is very convincing and of broad interest in the CRAC channel field...”*

1.1 *“In Figure 1 several engineered light-sensitive constructs are shown. Fig 1i shows light-sensitive constructs attached to the ER membrane. Another interesting approach would be if the authors link CRY2 via a TM-segment located in the ER-membrane to CCI-SOAR (in other words: CRY2 would localize in the lumen of the ER and be connected via a TM segment of CCI-SOAR located in the cytosol). This would even more mimic the wild-type STIM1 situation, just having a light-sensor instead of a Ca²⁺ sensor.”*

Response: We deeply appreciate the reviewer's constructive suggestion that ER lumen localized CRY2 connected via a TM segment of STIM1ct would more faithfully mimic the wild-type STIM1 situation. In our initial experiments, we indeed developed several CRY2-STIM1 variants with CRY2 within the ER lumen (e.g. SP_{ER}-mCh-CRY2_{PHR}-STIM1₃₈₋₆₈₅, SP_{ER}-mCh-CRY2_{PHR}-STIM1₁₂₈₋₆₈₅, and SP_{ER}-mCh-CRY2_{PHR}-STIM1₂₀₀₋₆₈₅; **Supplementary Fig. 2**). However, these CRY2-STIM1 hybrid proteins all failed to respond to blue light illumination. For example, SP_{ER}-mCh-CRY2_{PHR}-STIM1₃₈₋₆₈₅ failed to elicit Ca²⁺ influx or cause puncta formation even with overexpressed ORAI1 after 5 min photo-stimulation (**Supplementary Fig. 2b-2c**). However, it retained the ability to form puncta and co-cluster with ORAI1 after TG stimulation (**Supplementary Fig. 2c**), indicating that the hybrid STIM1 protein per se was still functional. This prompted us to speculate that the photosensory domain CRY2 would not work in a relatively oxidizing environment within the ER lumen, when compared to a more reducing cytosolic environment. To test this, we made two

CRY2 variants (using CRY2_{clust} because it tends to oligomerize more efficiently than the WT CRY2; PMID: 28646204): one within the ER lumen (SP_{ER}-mCh-CRY2_{clust}-KDEL) and the other anchored to ER membrane but with the CRY2 component facing toward the cytosolic side (SP_{ER}-mCh-TM-CRY2_{clust}). In the dark, both SP_{ER}-mCh-CRY2_{clust}-KDEL and SP_{ER}-mCh-TM-CRY2_{clust} showed even ER-like distribution. After blue light stimulation, only SP_{ER}-mCh-TM-CRY2_{clust} (with CRY2 facing the cytosol), but not the luminal version, formed clusters along ER tubules (**Supplementary Fig. 2a**). Clearly, CRY2 within the ER lumen cannot be photoactivated.

The N terminal PHR (Photolyase-Homologous Region, CRY2_{PHR}) domain of CRY2 binds non-covalently to the chromophore flavin adenine dinucleotide (FAD). Photoexcited CRY2 undergoes oligomerization or interacts with its partner (CIB1) via a photocycle or photon-driven redox exchanges between the flavin chromophore and its protein environment (PMID: 18988809, 17355959, 15774475 and 20943427). According to the photoreduction hypothesis, at the resting state (dark), CRY binds oxidized FAD (FAD_{ox}), which is reduced to semi-reduced FDAH• after blue light illumination. The FDAH• can be further reduced to the fully reduced (FADH⁻) form by blue light. The semi-reduced FDAH• acts as a major photoexcitation product that represents the signaling state of cryptochromes, which can be oxidized to regenerate the resting state oxidized flavin (FAD_{ox}) *via* dark conversion to complete the photocycle (PMID: 17355959, 15774475 and 20943427). The redox potential of FAD_{ox}/FADH• pair in *Arabidopsis* CRY was determined to be in the range of -143 to -153 mV (PMID: 7638620 and 19140781). The redox potential of an *Arabidopsis* cell is approximately -320 mV (PMID: 17892447 and 18778428), which is comparable to the cytoplasmic redox potentials of a typical mammalian cell (about -300 mV; PMID: 18469822 and 23242256). Compared with a more reduced condition in the cytoplasm, the endoplasmic reticulum (ER) lumen represents a more oxidized environment with redox potentials ranging from -118 mV to -208 mV measured by different methods (PMID: 1523409, 19026441, 22715429 and 23424194), which might prohibit the redox reaction to photoactivate CRY2. Indeed, it has been shown that CRY tends to be more easily activated with less light input in more reducing environment (PMID: 16164372). In addition, cryptochrome can be chemically reduced by a high concentration of reducing agent DTT to become catalytically active in vitro (PMID: 12797829 and 2059637). All these results converge to support the idea that a reducing environment is required to enable photoactivation of CRY. This may well explain our inability to activate CRY2 when resided within the ER lumen.

Taken together, we concluded that the blue light illumination cannot effectively reduce CRY2 in the more oxidized ER lumen and thus fail to photo-excite CRY2 (Redox potential: -118 mV in the ER lumen *versus* -300 mV in the cytosol). More engineering work has to be done to evolve CRY2 variants that might work in the ER lumen, but this will be beyond the scope of the current work.

1.2 “*CIBN-Cry2 and Cry2-constructs show slower reversibility than iLID-sspB constructs. The authors write that their studies suggest that store-refilling is sufficient for STIM1 oligomer to monomer transition. Which of the four approaches mimics most the physiological conditions, especially in respect to time-required for store-refilling/recovery from light-activated state and associated transition of STIM1 into the resting state?*”

Response: The kinetic parameters of using different approaches to reversibly photoactivate STIM1ct were summarized in **Supplementary Fig. 19a**. The ER store refilling process has been well studied by several groups using different reporters and agonists (PMID: 22464749, 21880734 and 17517596). The reported

half-lives of ER Ca²⁺ refilling, as listed in **Supplementary Fig. 19b**, largely fell in the range of 50 ~ 70 seconds. Indeed, this was consistent with our own measured half-life of the ER store refilling (~70 seconds) by using an ER Ca²⁺ sensor R-CEPIA_{ER} after carbachol stimulation (PMID: 29934936). With regard to STIM1 oligomerization and deoligomerization, a FRET pair made of CFP or YFP-fused to the luminal side of STIM1 reported a half-life of about ~26 seconds for activation and ~21 seconds for de-oligomerization (PMID: 17517596). By taking into account these values, we think iLID/sspB fused STIM1ct ($t_{1/2, \text{on}} = 28.5 \pm 3.2 \text{ sec}$ and $t_{1/2, \text{off}} = 48.6 \pm 5.4 \text{ sec}$) may most closely mimic the physiological conditions.

1.3 “*Supp Fig 1: Can you please provide a time course for NFAT translocation.?*”

Response: We have added the time course of NFAT nuclear translocation in the revised **Supplementary Fig. 3d**.

1.4 “*Cry2 and LOV2 attachment to STIM1 fragments have been performed by some groups. Can you please highlight and reference that and of who me basic principles have already been published.*”

Response: We have followed the advice to highlight and reference all existing STIM1-based optogenetic tools and briefly discussed the basic principles in the beginning of the Discussion section: “*The prototypical CRAC channel made of ORAI1 and STIM1 serves as a major route for Ca²⁺ entry in many cell types. Various optogenetic modules have been engineered into STIM1 to confer light-sensitivity to CRAC channel over the past five years^{5, 6, 41-47}. We call these tools as genetically encoded Ca²⁺ actuators (GECAs)⁶, as opposed to genetically-encoded Ca²⁺ indicators (GECIs) that are widely used for monitoring Ca²⁺ signals⁴⁸. Two major strategies have been adopted to make GECAs: fusion with CRY2 to mimic Ca²⁺-depletion induced STIM1 oligomerization or replacing CC1 with LOV2 to recapitulate intramolecular autoinhibition with the STIM1 cytoplasmic domain^{5, 6, 41-47}. Herein, by using STIM1 as a test case, we have further exploited these optogenetic engineering approaches to study protein activities, interrogate and control cell signaling.*”

1.5 “*In Figure 3 the authors present the series of CCI truncation mutants showing that CCI-L251-L261 is critical for CCI-SOAR interaction. The enhance the reliability of this assay: Can the authors also show that truncation of part of SOAR abolishes Ca²⁺ entry, independent of the dark or light state?*”

Response: We have followed the reviewer’s suggestion to truncate part of SOAR (aa 344-442) using two constructs: mCh-CRY2-STIM1₂₃₃₋₄₀₀ and mCh-CRY2-STIM1₂₃₃₋₄₃₀. Both constructs failed to induce Ca²⁺ influx in the dark or under blue light stimulation (**Supplementary Fig. 10**), thereby attesting to the notion that the structural integrity of SOAR is required for optogenetic activation of CRAC channels.

1.6 “*Regarding the prey and bait approach in Figure 4: The authors show that the CRY2-EF domain enables light-induced clustering, but no interaction with other EF-domains. What is the effect of luminal Ca²⁺ in this case? This approach seems to be done in the cytosol with low Ca²⁺ levels.*”

Response: To evaluate the effect of Ca^{2+} , we co-expressed mCh-CRY2-EF and YFP-EF domains in HeLa cells and then treated the cells with digitonin, which can permeabilize cell membrane to enable Ca^{2+} exchange between the cytoplasm and the extracellular medium ($[\text{Ca}^{2+}]$: 1.8 mM). We confirmed the rapid increase of cytosolic Ca^{2+} upon addition of 5 μM digitonin to the cells, but did not observe the co-clustering between mCh-CRY2-EF (construct B4) and YFP-EF (construct P6) after photo-stimulation in the presence of either low (resting cytosolic Ca^{2+}) or high (mM) Ca^{2+} (**Supplementary Fig. 11d**). These results suggest that, regardless of high Ca^{2+} , the EF-hands have no preference to heterodimerize.

1.7 *“Another interesting approach would be if the authors attach Cry2 at the NT of full-length STIM1 and investigate the clustering efficiency in dependence of the absence or presence of EF, SAM, ...”*

Response: We thank the reviewer for this constructive suggestion. We made the recommended constructs. However, CRY2 showed loss-of-function in ER lumen due to a highly oxidized environment, thus preventing us to perform the suggested experiments. For detailed explanation, please refer to our response to Comment 1.1 and also see **Supplementary Fig. 2**.

1.8 *“The authors demonstrate a predominant localization of mCh-LOV2-SOAR to the ER than to the PM upon illumination with light and in the presence of STIM1 1-342 and Orai1 at a ratio of 1:1. What would happen if the ratio would be altered either via additional overexpression of STIM1 1-342 or Orai1? This could show if the ratio plays a role or not. Furthermore the 1:1 ratio is probably not physiological.”*

Response: To better estimate the ‘tug-of-war’ between CC1 (in the context of ER-resident STIM1₁₋₃₄₂) and ORAI1 to engage the SOAR domain, we co-expressed uncaged SOAR/CAD domain (mCh-CAD) with different ratios of STIM1₁₋₃₄₂-CFP and YFP-ORAI1 in HeLa cells (**Supplementary Fig. 6**). In HeLa cells over-expressing STIM1₁₋₃₄₂-CFP or YFP-ORAI1, mCh-CAD predominantly located at ER or PM, respectively (**Supplementary Fig. 6b-d**). In HeLa cells expressing STIM1₁₋₃₄₂ and ORAI1 at a ratio of ~1:1 (**Supplementary Fig. 6a**), mCh-CAD was primarily anchored toward the ER membrane, displaying an ER-like tubular distribution pattern at the resting condition (**Supplementary Fig. 6a**, top panel). After TG-induced store depletion, mCh-CAD was more dispersed into the cytosol and showed partial co-localization with PM-embedded ORAI1 (**Supplementary Fig. 6a**, bottom panel). In cells with the ratio of STIM1₁₋₃₄₂ to ORAI1 less than 1:1, mCh-CAD exhibited an ER-like distribution but with partial PM decoration (**Supplementary Fig. 6b and 6d**). The excessive expression of ORAI1 indeed recruited a portion of mCh-CAD toward PM. Taken together, it is clear that SOAR/CAD has a relatively higher binding affinity toward CC1 than to ORAI1. Under physiological scenario, we agree with the reviewer that the 1:1 ratio is probably not physiological, and that the ratio of CC1 and ORAI1 will affect SOAR localization. Extrapolating this finding to certain pathological conditions where ORAI1 was abnormally upregulated (PMID: 26017146, PMID: 24954132), we believe that the CRAC channel activation and deactivation kinetics will likely be altered.

1.9 *“The authors describe in detail the effects of diverse single point mutations at position T393. However, the relevance of 393 remains unclear. Is it a disease related mutant? Why were other mutations not investigated in such detailed manner? Further T393 is close to the critical and prominent F394 mutant, which has not been*

evaluated in Fig 5c.”

Response: Point mutation at position T393 was not detected in the current TCGA data portal, nor in the COSMIC database of somatic mutations in cancer. T393 is strategically located at the turn between CC2 (SOAR- α 1) and CC3 (SOAR- α 2-4) based on the SOAR crystal structure. T393 is also very close to the proposed ORAI binding region (³⁸²KIKKK³⁸⁶) and S α 3 (aa 400-403). In the current study, to our surprise, a series of T393X mutations showed variegated outcomes in terms of Ca²⁺ influx (indirect readout for ORAI gating) and PM localization (indirect readout for ORAI binding), as shown in **Fig 5c-i**. Hence, exploring T393 mutations might help us to better understand how the SOAR apical region affects ORAI binding and channel gating. This is the major reason driving us to single out T393x mutations in the study.

Regarding the position of F394, systematic mutational studies (F394-L/A/H) in the context of SOAR domain and the full length STIM1 have been nicely done by the Gill group over the past years (PMID: 24492416, 26399906, 29581306). In our opinion, simply repeating their studies using a different approach might be deemed as an incremental advance to the field. Thus, we decided to focus on T393 mutations that have not been addressed in previous structure-function relationship studies.

1.10 *“The assay in Fig5 e/g shows mutants with very diverse functional effects. T393F shows no Ca²⁺ entry, but strong PM localization independent of dark or light state, suggesting a defect in gating. T393P shows light-induced Ca²⁺ entry but very low PM localization – how can this be explained? I383K shows no Ca²⁺ entry, but light-induce pm-localization – is this another gating deficient effect?”*

Response: We agree with the reviewer that the T393F mutation compromised the channel gating ability. T393P showed less prominent PM localization but retained the ability to gate ORAI channels, suggesting it might represent a more potent ORAI activator. For the mutant I383K in the context of CRY2-STIM1ct, we observed a light-dependent translocation from the cytosol to the plasma membrane, but it failed to elicit Ca²⁺ influx. When we introduced the mutation into the full-length STIM1, we observed a constitutive formation of STIM1 puncta at ER-PM junctions in ORAI-null HEK293 cells at rest, suggesting that the ³⁸²KIKKK³⁸⁶>KIKKK substitution at the SOAR apex region might promote the interaction between STIM1 and PM-resident negatively charged phosphoinositides as the polybasic C-tail does. We have added more discussion in the main text to explain the diverse functional effects of T393 mutations.

1.12 *“Figure 5 c includes a series of unlabeled dots. Which mutants do they represent? Moreover, the WT dot: does it represent the mCh-cry2-STIM1ct wt construct?.. Also Fig 5j shows several dots without labelling. Please add labels or remove dots.”*

Response: We have labeled all the dots in **Fig. 5**. We added the note for WT, which represented the mCh-CRY2-STIM1ct construct.

1.13 *“Supp Fig 7/8: Why are some mutants encircled via red rectangles?”*

Response: The red color was intended to highlight the mutants. To avoid misunderstanding, we deleted red rectangles in all supplementary figures.

1.14 “Also the MT/PB-approach is very impressive. Still the question remains if a Cry2-STIM1-CT construct behaves in the same manner regarding localization to microtubule. Maybe the authors could provide some key experiments with their Cry2-S1-CT construct showing in the w.t. conditions and upon P645N and PL/KK mutation the MT and PM association.”

Response: Following the reviewer’s recommendation, we compared the behaviors of CRY2-STIM1ct WT and the mutants (P645N, PL/KK, and P645N+PL/KK) in **Supplementary Fig. 18**. After blue light illumination, CRY2-STIM1ct WT displayed comet-like movements by tracking along microtubule (MT) and also elicited Ca²⁺ entry (**Supplementary Fig. 18a, c**). CRY2-STIM1ct did not show overt cytosol-to-PM translocation with the endogenous level of ORAI1 in HeLa cells. When the +TIP-disrupting mutation P645N was introduced, CRY2-STIM1ct failed to track MT plus ends but retained the ability to photo-activate Ca²⁺ influx, which showed a slightly slower kinetics compared to WT ($t_{1/2,on}$: 56.6 vs 44.2 sec; **Supplementary Fig. 18c**). The CRY2-STIM1ct (P645N+PL/KK) mutant, which was designed to enhance STIM1-phospholipid interaction, showed faster light-dependent cytosol-to-PM translocation and Ca²⁺ influx (**Supplementary Fig. 18c**). These data suggested that both the S/TxIP motif the PB domains can facilitate more rapid and efficient targeting of STIM1 toward the PM.

1.15 “The authors reported that LOV2-SOAR preferentially goes to the ER than to the PM under 1:1 S1:O1 ratio. They suggest that further domains are required for preferential recruitment to the PM. So what happens if the attached LOV2 to a fragment 344-685, thus including SOAR, TRIP and PB domains? Mutations in TRIP and PB could narrow down potential domains which allow preferential recruitment to the PM.”

Response: We have followed the reviewer’s valuable suggestion and made the construct mCh-LOV2-STIM1₃₃₆₋₆₈₅. Light inducible cytosol-to-PM translocation of mCh-LOV2-STIM1₃₃₆₋₆₈₅ was first evaluated in HeLa cells with endogenous ORAI. Given the low endogenous ORAI levels, we did not observe notable PM translocation of mCh-LOV2-STIM1₃₃₆₋₆₈₅ (**Supplementary Fig. 5a**). Next, we overexpressed YFP-ORAI in HeLa cells and did the same experiment. We detected light-inducible cytosol-to-PM translocation of mCh-LOV2-STIM1₃₃₆₋₆₈₅ (**Supplementary Fig. 5b**). Worth of noting here is that, even in the dark, mCh-LOV2-STIM1₃₃₆₋₆₈₅ partially co-localized with YFP-ORAI1 in the PM, suggesting that LOV2 could not fully cage STIM1₃₃₆₋₆₈₅. This is understandable given that we have devoted tremendous efforts (hundreds of constructs) to optimize the linker and fragments of STIM1ct to make LOV2-STIM1₃₃₆₋₄₈₆ effectively caged by LOV2. Investing more time to optimize LOV2-STIM1₃₃₆₋₆₈₄ apparently is beyond the scope of the current study.

Regardless, a side-by-side comparison between mCh-LOV2-STIM1₃₃₆₋₄₈₆ and mCh-LOV2-STIM1₃₃₆₋₆₈₅ was performed with the results shown in **Supplementary Fig. 5d-f**. First, compared with LOV2-STIM1₃₃₆₋₄₈₆, LOV2 could not fully cage STIM1₃₃₆₋₆₈₅, resulting in higher Ca²⁺ background (**Supplementary Fig. 5d-e**) and partial ORAI1 binding in some cells even in the dark (**Supplementary Fig. 5b-c**). Second, with respect to CC1-SOAR interaction *in trans*, unlike LOV2-STIM1₃₃₆₋₄₈₆, LOV2-STIM1₃₃₆₋₆₈₅ did not show overt docking toward ER-resident STIM1₁₋₃₄₂ before and after light illumination. Indeed, other structural elements downstream of the SOAR domain, such as the TRIP and PB motifs, likely exert additional forces to promote

STIM1₃₃₆₋₆₈₅ moving toward the PM. Mutations in TRIP and PB have been systematically evaluated in **Fig. 6** and **Supplementary Figs. 17-18**.

1.16 “Minor: Page 4: progress instead of prowess.”

Response: Corrected as suggested.

Reviewer #2

We are pleased that the reviewer describes that our paper “Overall, this technically rigorous and careful study highlights an interesting new direction for the use of optogenetic actuators.”

2.1 “The authors could provide a more thorough quantification of their live-cell imaging data. Currently, only a subset of experiments are presented with quantification, mostly via calcium indicator time-courses or changes in calcium release or localization, while many results are summarized using representative images alone (e.g., Figure 4h, Supplementary Figures 3-5). Notably, very few time-courses directly tracking the relocalization/redistribution of tagged STIM1 constructs in response to illumination are shown, though it would be useful to compare the kinetics of optogenetic switching with downstream events (e.g., calcium release in Figure 1).”

Response: Following the reviewer’s recommendation, we have performed quantitative analysis in **Fig. 4h**, **Supplementary Fig. 3-5** (now **Supplementary Figs. 7-9**) and other newly added figures. Please see the bar graphs shown in the related figures. The quantitative data for **Fig. 4h** was shown in **Supplementary Fig. 11**.

The time course for photo-induced Ca²⁺ influx and redistribution of optogenetics engineered STIM1 constructs were shown in **Supplementary Fig. 1** and **Supplementary Fig. 18** for CRY2/CIBN-STIM1ct variants and in **Supplementary Fig. 5** for LOV2 caged STIM1ct fragments. Please see these revised figures for comparison of kinetics.

2.2 “Looking at Supplementary Figure 10 and Supplementary Video 5, the mCh-CRY2-PB (PL/KK) construct appears to exhibit fairly strong nuclear localization at rest and to undergo striking and dynamic redistribution within the nucleus upon illumination. However, this behavior is not mentioned in the text. Could the authors please include some comment on what might be happening? Is this merely an artifact?”

Response: We thank the reviewer for pointing out this. For the construct of mCh-CRY2-PB, the PB domain is derived from aa 671-685 of STIM1 (RKKFPLKIFKKPLKK), which contains multiple stretches of positively-charged amino acids. This feature is very similar to a typical nuclear localization signal (NLS) that often bears a cluster of basic residues (e.g. PKKKRKV in the SV40 Large T-antigen; or AVKRPAATKKAGQAKKKKLD in nucleoplasmin). In our study, we further introduced two additional lysine residues (PL>KK) to enhance its lipid binding capability, while undesirably boosting its tendency for nuclear localization. To the contrary, after introducing the K684A mutation into PB, we observed a reduction of PM and nuclear localization of CRY2-PB (**Supplementary Fig. 17b**; second panel). We have added the following sentence in the related figure legend to explain this phenomenon: “Because the PB domain resembles nuclear localization

sequences that are rich in positively-charged residues, CRY2-PB variants showed both PM-like and nuclear localization.”

2.3 *“The sentence on page 5 starting with “Furthermore, this photo-induced process can be readily reversed...” should be clarified. From the context, it seems like the authors may be referring to spontaneous dissociation of STIM and Orai being sufficient to drive reversal in the absence of pro-dimerization signals, however the wording is somewhat awkward.*

Response: We have revised our text shown as follows: *“Furthermore, upon the withdrawal of light stimulation, the optogenetic module undergoes dissociation, a process that resembles the binding of Ca²⁺ to the luminal EF-SAM domain to drive the de-oligomerization of STIM1 upon ER Ca²⁺ store refilling^{13, 25}. In the absence of pro-oligomerization signals, STIM1ct will dissociate from ORAI and adopt a folded-back conformation via intramolecular CC1-SOAR trapping. We speculate that forced separation of the juxtamembrane ends of CC1 might be sufficient to bring STIM1 back to its resting configuration, even in the absence of other ancillary proteins.”*

Other points:

2.4 *“On page 10, paragraph 2, “Fig 3f-g” and “Fig 3d-e” should refer to Figure 5.”*

Response: Corrected.

2.5 *“At the top of page 11, “CRY2-baesda” should be “CRY2-based”.”*

Response: Corrected.

2.6 *“At the top of page 12, “SAOR” should be “SOAR”.”*

Response: Corrected.

2.7 *“In cartoon in Figure 2c, the LOV and SOAR domains switch colors after light stimulation. Also, in Figure 2f, the mCh channel images are show in grayscale but are described as “red” in the legend.”*

Response: Corrected.

2.8 *“The legends for Figure 5e and f are inverted with respect to what is shown in the figure.”*

Response: Corrected.

2.9 *“On the last page of the Supplement, “Supplementary Videos 1-4” should be “1-5”.”*

Response: Corrected.

REVIEWERS' COMMENTS:

Reviewer #1 (Remarks to the Author):

No further comments.

Reviewer #2 (Remarks to the Author):

The authors have done an excellent job of addressing the majority of the points raised by reviewers.

However, I don't see any new quantification for Supplementary Fig. 8 (previously Supplementary Fig. 4). The other added quantification looks good.

Also, the authors have not corrected the cartoon in Fig. 2C - the colors of the LOV2 and SOAR domains still switch from red LOV2/green SOAR on the left-hand side to green LOV2/red SOAR on the right-hand side.

Reviewer #1

No further comments.

Reviewer #2

1. The authors have done an excellent job of addressing the majority of the points raised by reviewers. However, I don't see any new quantification for Supplementary F/ig. 8 (previously Supplementary Fig. 4). The other added quantification looks good.

Response: We provided the quantification data for Supplementary Fig. 8.

2. Also, the authors have not corrected the cartoon in Fig. 2C - the colors of the LOV2 and SOAR domains still switch from red LOV2/green SOAR on the left-hand side to green LOV2/red SOAR on the right-hand side.

Response: We have corrected the color code in the cartoon as advised.